# Challenge of modelling GLORIA observations of UT/LMS trace gas and cloud distributions at high latitudes: a case study with state-of-the-art models

Florian Haenel[1], Wolfgang Woiwode[1], Jennifer Buchmüller[2], Felix Friedl-Vallon[1], Michael Höpfner[1], Sören Johansson[1], Farahnaz Khosrawi[1], Oliver Kirner[2], Anne Kleinert[1], Hermann Oelhaf[1], Johannes Orphal[1], Roland Ruhnke[1], Björn-Martin Sinnhuber[1], Jörn Ungermann[3], Michael Weimer[1,2,*] and Peter Braesicke[1]

[1]Institute of Meteorology and Climate Research, Karlsruhe Institute of Technology, Karlsruhe, Germany
[2]Steinbuch Centre for Computing, Karlsruhe Institute of Technology, Karlsruhe, Germany
[3]Institute of Energy and Climate Research – Stratosphere (IEK-7), Forschungszentrum Jülich, Jülich, Germany
[*]now at: Department of Earth, Atmospheric and Planetary Sciences, Massachusetts Institute of Technology, Cambridge, MA, USA

*Correspondence to*: Florian Haenel (florian.haenel@kit.edu)

**Abstract.** Water vapour and ozone are important for the thermal and radiative balance of the upper troposphere (UT) and lowermost stratosphere (LMS). Both species are modulated by transport processes. Chemical and microphysical processes affect them differently. Thus, representing the different processes and their interactions is a challenging task for dynamical cores, chemical modules and microphysical parameterisations of state-of-the-art atmospheric model components. To test and improve the models, high resolution measurements of the UT/LMS are required. Here, we use measurements taken in a flight of the GLORIA (Gimballed Limb Observer for Radiance Imaging of the Atmosphere) instrument on HALO. The German research aircraft HALO (High Altitude and LOng range research aircraft) performed a research flight on 26 February 2016 that covered deeply subsided air masses of the aged 2015/16 Arctic vortex, high-latitude LMS air masses, a highly textured region affected by troposphere-to-stratosphere exchange, and high-altitude cirrus clouds. Therefore, it provides a challenging multifaceted case study for comparing GLORIA observations with state-of-the-art atmospheric model simulations in a complex UT/LMS region at a late stage of the Arctic winter 2015/16.

Using GLORIA observations in this manifold scenario, we test the ability of the numerical weather prediction (NWP)-model ICON (ICOsahedral Nonhydrostatic) with the extension ART (Aerosols and Reactive Trace gases) and the chemistry-climate model (CCM) EMAC (ECHAM5/MESSy Atmospheric Chemistry) to model the UT/LMS composition of water vapour ($H_2O$), ozone ($O_3$), nitric acid ($HNO_3$) and clouds. Within the scales resolved by the respective model, we find good overall agreement of both models with GLORIA. The applied high-resolution ICON-ART setup involving a R2B7 nest (local grid refinement with a horizontal resolution of about 20 km), covering the HALO flight region, reproduces mesoscale dynamical structures well. Narrow moist filaments in the LMS observed by GLORIA at tropopause gradients in context of a Rossby wave breaking event and in the vicinity of an occluded Icelandic low are clearly reproduced by the model. Using ICON-ART, we show that

a larger filament in the west was transported horizontally into the Arctic LMS in connection with a jet stream split associated with poleward breaking of a cyclonically sheared Rossby wave. Further weaker filaments are associated with an older tropopause fold in the east. Given the lower resolution (T106) of the nudged simulation of the EMAC model, we find that this model also reproduces these features well. Overall, trace gas mixing ratios simulated by both models are in a realistic range,

and major cloud systems observed by GLORIA are mostly reproduced. However, we find both models to be affected by a well-known systematic moist bias in the LMS. Further biases are diagnosed in the ICON-ART $O_3$, EMAC $H_2O$ and EMAC $HNO_3$ distributions. Finally, we use sensitivity simulations to investigate (i) short-term cirrus cloud impacts on the $H_2O$ distribution (ICON-ART), (ii) the overall impact of polar winter chemistry and microphysical processing on $O_3$ and $HNO_3$ (ICON-ART/EMAC), (iii) the impact of the model resolution on simulated parameters (EMAC), and (iv) consequences of

scavenging processes by cloud particles (EMAC). We find that changing the horizontal model resolution results in notable systematic changes for all species in the LMS, while scavenging processes play a role only in case of $HNO_3$. We discuss the model biases and deficits found in this case study that potentially affect forecasts and projections (adversely), and provide suggestions for further model improvements.

## 1 Introduction

Trace gas composition, in particular the vertical distributions of greenhouse gases, and clouds play an important role in the thermal and radiative budget of the upper troposphere/lowermost stratosphere (UT/LMS) (e.g. Riese et al., 2012; Hartmann et al., 2013). Stratospheric and, particularly, lowermost stratospheric water vapour has been identified to be an important driver in decadal global surface climate change (e.g. Forster and Shine, 2002; Solomon et al., 2010). Also, changes in stratospheric ozone are well known to affect temperature trends and radiative forcing (e.g. Forster and Shine, 1997). In the lower

stratosphere, ozone depletion is a major contributor to its negative temperature trend. There is also a significant spread among modelled trends when ozone and other greenhouse gas abundances are perturbed. Explanations for such differences include the different responses of individual radiation schemes and different sensitivities in the dynamical forcing in the models to changes in trace gases (e.g. Shine et al., 2003). Lowermost stratospheric water vapour distributions show hemispheric differences, thus requiring knowledge of hemispheric and latitudinal distributions and change for accurate climate projections

(e.g. Kelly et al., 1991; Rosenlof et al., 1997; Pan et al., 1997).

The LMS is the lowest layer of the stratosphere situated between the local tropopause and the 380 K isentropic level (e.g. Werner et al., 2010). In the winter hemisphere, its composition is mainly affected by air mass contributions from the polar winter vortex, the mid-latitude stratosphere, and the troposphere. While air masses in the polar winter vortex are mostly isolated from the surrounding stratosphere, LMS air masses at the bottom of the polar vortex can be affected significantly by

interactions with air masses from lower latitudes (e.g. Krause et al., 2018).

Rossby waves are undulations of the eastward-directed upper-tropospheric flow in the midlatitudes and are accompanied by step-like changes in the height of the dynamical tropopause (e.g. Wirth et al., 2018). Rossby wave breaking events can be

identified as overturning patterns in Ertel's potential vorticity (PV) and contribute to exchange of upper tropospheric and lower stratospheric air masses (e.g. Gabriel and Peters, 2008; Jing et al., 2018).

Exchange processes including quasi-isentropic and cross-isentropic exchange occur often in the vicinity of jet streams (e.g. Holton et al., 1995; Gettelman et al., 2011). They can be accompanied by different kinds of tropopause folds and modulate the

trace gas composition of the UT/LMS. Irreversible fluxes between the UT and the LMS can occur in either direction – from stratosphere-to-troposphere and from troposphere-to-stratosphere. Generally, the dominating flux in the extratropics is directed towards the troposphere. Such exchange processes and their effects have been investigated by numerous field observations (e.g. Ray et al., 1999; Hoor et al., 2002, 2005; Bönisch et al., 2009; Krause et al., 2018) and by many theoretical and modelling studies (e.g. Meloen et al., 2003; Stohl et al., 2003 and references therein).

Cirrus clouds are one of the least understood factors modulating climate change and affecting the composition of the UT/LMS (e.g. Schiller et al., 2008; Barahona and Nenes, 2009). Cirrus clouds absorb upwelling infrared light and reflect sunlight back to space and thereby affect the radiative budget and thus the thermal structure of the tropopause region. Sedimentation of cirrus cloud ice particles redistributes water vertically and changes the water vapour profile. Furthermore, the ice particles are capable of trapping nitric acid and other trace gases (e.g. Popp et al., 2004; Voigt et al., 2006; Krämer et al., 2008; Kärcher et al., 2009).

Moreover, vertical distributions of $H_2O$ and $HNO_3$ altered by cirrus cloud processing might affect the availability of reactive nitrogen oxides ($NO_x$) and hydroxyl radicals, which are again important factors affecting the local concentrations of ozone and methane (e.g. Kelly et al., 1991; Krämer et al., 2008; Schiller et al., 2008).

Nowadays, numerical weather prediction and chemistry-climate models (NWPs and CCMs) are capable of resolving the UT/LMS, mesoscale dynamics and cloud processes in part explicitly and in part by using parameterisations ranging from low

to high complexity. Examples of such models include ICON (ICOsahedral Nonhydrostatic, see Zängl et al., 2015) with the extension ART (Aerosols and Reactive Trace gases, see Rieger et al., 2015 and Schröter et al., 2018) and EMAC (ECHAM5/MESSy Atmospheric Chemistry, see Jöckel et al., 2006, 2010, 2015 and Roeckner et al., 2006). However, accurate simulations of UT/LMS composition, dynamics and cirrus clouds (and their interactions) remain a challenge and are important building blocks for reliable weather forecasting and climate projections. In particular, LMS water vapour is known to be

affected by significant systematic errors in model simulations (e.g. Stenke et al., 2008).

The exceptionally cold Arctic winter 2015/16 was characterised by a stable polar vortex and low temperatures in the UT/LMS region (Matthias et al., 2016). While the winter was the coldest on record from December to early February, complex dynamical processes and a major final stratospheric warming in early March ended the cold phase and resulted in a vortex split in mid-March (Manney and Lawrence, 2016). In the same winter, airborne observations in the framework of the combined

POLSTRACC (POLar STRAtosphere in a Changing Climate), GW-LCYCLE (Gravity Wave Life Cycle Experiment) II and SALSA (Seasonality of Air mass transport and origin in the Lowermost Stratosphere using the HALO Aircraft) (PGS) field campaign probed the Arctic UT/LMS region in the period from December 2015 to March 2016 (Oelhaf et al., 2019). During PGS, the GLORIA (Gimballed Limb Observer for Radiance Imaging of the Atmosphere) instrument (Friedl-Vallon et al., 2014; Riese et al., 2014) was deployed on-board the German HALO (High Altitude and LOng Range Research Aircraft). From

the GLORIA limb-imaging observations, vertical distributions of temperature, trace gases and clouds are derived and allow detailed model comparisons (e.g. Khosrawi et al., 2017; Braun et al., 2019; Johansson et al., 2019).

During the research flight on 26 February 2016 (PGS 14), GLORIA probed subsided LMS air masses of the aged 2015/16 polar vortex in high latitudes, a highly textured region affected by troposphere-stratosphere exchange, and high-altitude cirrus

clouds across a long transect spanning from Scandinavia over Greenland to Canada. Here, we use the GLORIA observations during this flight to test the capabilities of EMAC and ICON-ART of modelling mesoscale $H_2O$, $O_3$ and $HNO_3$ distributions and cirrus clouds and to reveal discrepancies and deviations that might be related to (systematic) biases in the modelled trace gas distributions. We particularly focus on a troposphere-stratosphere exchange region in the vicinity of an occluded Icelandic low. Finally, we use sensitivity simulations to investigate (i) short-term cirrus cloud impacts on the $H_2O$-distribution (ICON-

ART), (ii) the impact of polar winter chemistry and microphysical processing on $O_3$ and $HNO_3$ (ICON-ART/EMAC), (iii) the impact of model resolution on simulated parameters (EMAC), and (iv) consequences of scavenging processes by cloud particles (EMAC).

In Section 2, we introduce our observations, models and diagnostics. An overview of the meteorological situation and the GLORIA observations during PGS 14 is provided in Sect. 3. In Section 4, the 2-dimensional vertical cross sections of modelled

cloud and trace gas distributions are compared with the GLORIA observations, discrepancies are diagnosed and investigated, and sensitivity experiments with the models are presented. We furthermore investigate the evolution of narrow moist filaments observed by GLORIA in the LMS with the aid of ICON-ART. The results are summarised and discussed in Sect. 5.

## 2 Data and diagnostics

In the following, the characteristics of the GLORIA observations, the model setups used, and the applied diagnostics are

introduced. An overview of the cloud and trace gas products used is provided in Tables 1 and 2.

### 2.1 GLORIA observations

The GLORIA data used here were measured during the HALO flight PGS 14 on 26 February 2016. PGS 14 started in Kiruna, northern Sweden, and covered the Arctic Sea, Greenland, and Eastern Canada (Fig. 1b). GLORIA is a passive infrared limb-imaging spectrometer deployed on-board high-altitude aircraft (Friedl-Vallon et al., 2014; Riese et al., 2014). GLORIA uses

128 vertical times 48 horizontal pixels of a Mercury Cadmium Telluride (HgCdTe) detector coupled to an interferometer to measure thermal radiation of the atmosphere across the limb (Fig. 1a). The line-of-sight of GLORIA is actively controlled and stabilised by a gimballed frame. GLORIA covers a spectral range from 780 cm$^{-1}$ to 1400 cm$^{-1}$. Here, we use observations in the high spectral resolution mode (called "chemistry mode"), which involves a spectral sampling of 0.0625 cm$^{-1}$. In "chemistry

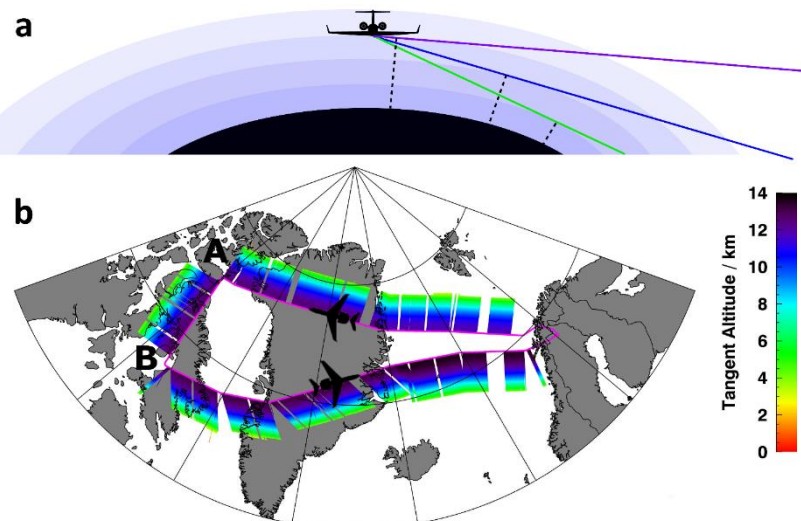

**Figure 1.** (a) Schematic representation of airborne limb viewing geometry. (b) GLORIA sampling during PGS14 on 26 February 2016. The tangent points of the GLORIA limb views are colour-coded with altitude. Characteristic waypoints are marked (A, B).

mode", one data cube is recorded within ~13 s (~3 km along flight track) and covers 128 vertical angles from ~5 km to flight altitude plus upward viewing angles simultaneously. Within one data cube, spectra of pixel rows are binned to reduce the noise. From the binned and calibrated spectra, vertical profiles of atmospheric parameters are derived. Thereby, one complete set of atmospheric parameter profiles (i.e. temperature, trace gases and cloud parameters) is obtained from one single data

cube. For each atmospheric parameter, the obtained profiles are combined into a 2-dimensional time-height cross section along the flight track.

Optical information on vertical cloud coverage is obtained directly from the calibrated spectra by using the cloud index method (Spang et al., 2004). The cloud index uses the colour ratio between the spectral microwindows from 788.20 cm$^{-1}$ to 796.25 cm$^{-1}$ and 832.30 cm$^{-1}$ to 834.40 cm$^{-1}$. Details on the applied 1-D trace gas retrieval and the data products used here are provided

by Johansson et al. (2018a). The retrieved individual trace gas profiles of GLORIA are combined to 2-D vertical cross sections of the respective species along the flight track. In the gas-phase $H_2O$ retrieval, one spectral transition in the microwindow from 795.7 cm$^{-1}$ to 796.1 cm$^{-1}$ is used. $O_3$ is retrieved using the spectral microwindows from 780.6 cm$^{-1}$ to 781.7 cm$^{-1}$ and 787.0 cm$^{-1}$ to 787.6 cm$^{-1}$. Gas-phase $HNO_3$ is retrieved using the spectral microwindows from 862.0 cm$^{-1}$ to 863.5 cm$^{-1}$, 866.1 cm$^{-1}$ to 867.5 cm$^{-1}$, and 901.3 cm$^{-1}$ to 901.8 cm$^{-1}$. As the retrieval of trace gases is not possible in the presence of optically thick clouds,

GLORIA limb spectra have been filtered by a dedicated cloud filter based on the cloud index. The estimated accuracy of the GLORIA data amounts to 10 % to 20 % for the respective trace gases (Johansson et al., 2018a). Typical vertical resolutions between 300 and 700 m are achieved for these trace gases.

## 2.2 ICON-ART chemistry-transport simulations

The state-of-the-art global meteorological forecast system ICON (Zängl et al., 2015) has been operational at the German Weather Service (Deutscher Wetterdienst, DWD) since 2015. ICON is developed by the DWD in cooperation with the Max-Planck-Institute for Meteorology, Hamburg. ICON uses a triangular grid, which is well suited for modern computer
architectures. Further, it allows efficient scaling of the dynamical core, avoids meridional grid-convergence and singularities

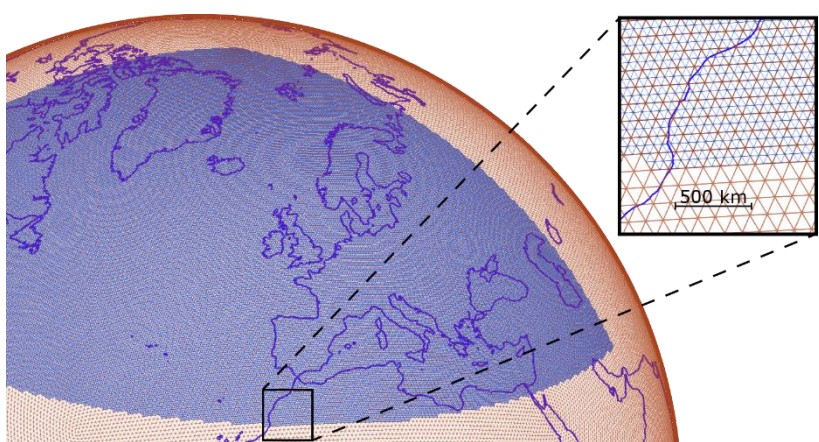

**Figure 2.** For the POLSTRACC winter, a global ICON-ART simulation with a global R2B6 grid was carried out (red). In the area of the flights, a nest with a R2B7 grid with ~20 km horizontal grid spacing was used to resolve mesoscale processes in more detail (blue).

at the poles, improves mass conservation and allows efficient local grid refinement with two-way interaction (nesting). In the vertical domain, a hybrid height coordinate is used (Leuenberger et al., 2010) that continuously transforms from local topography-following levels to constant height levels at 16 km and above.

The Aerosols and Reactive Trace gases module ART is developed at the Karlsruhe Institute of Technology (KIT). It simulates
chemical processes and aerosols, and couples trace gas concentrations and aerosols at each model time step to other relevant processes (Rieger et al., 2015; Schröter et al., 2018). The ICON transport scheme redistributes the tracers, and clouds and radiation properties are coupled to the meteorological state. ART is capable of simulating chemical and photo-chemical production and loss of reactive trace gases and can be used with defined emission scenarios (Weimer et al., 2017).

For the PGS campaign, a dedicated ICON-ART simulation was performed for the entire polar winter 2015/16 using a R2B6
(~40 km horizontal grid spacing) global grid. In the focus region around Scandinavia and Greenland, a R2B7 nest with a horizontal grid spacing of 20 km was applied (Fig. 2). The potential of the nesting property was recently shown by Weimer et al. (2021). In the vertical, 90 model levels from the ground to 75 km were employed, corresponding to a vertical resolution of ~400 m in the vertical region of interest here. Concerning the meteorology, the simulation was set up in a constrained forecast mode. Every day at 0 UTC, the atmospheric state (pressure, temperature, wind, potential vorticity (PV), as well as specific
humidity ($q_v$), and cloud parameters) was reinitialised using operational ECMWF (European Centre for Medium-Range Weather Forecasts) Integrated Forecast System (IFS) data at horizontal resolution of T1279 (approx. 16 km) and with 137 vertical levels (see Ehard et al., 2018). Therefore, small discontinuities in the meteorological state (including $q_v$) are possible

at the reinitialisation points. To investigate cirrus cloud effects on the LMS water vapour distribution on short forecast time scales, we furthermore use the tracer "$H_2O$ passive". This tracer is mostly identical with $q_v$ (including regular reinitialisation at 0 UTC), but does not account for cloud microphysics (i.e. nucleation and sedimentation of ice particles).

Other than the meteorological variables, tracers, such as the ozone tracers, are simulated continuously in a free-running mode after initialisation at the beginning of the winter using a previous EMAC simulation (Schröter et al., 2018) and are not reinitialised regularly at 0 UTC. The simulation of polar stratospheric ozone loss in the simulated "$O_3$ tracer" was done using linearised ozone chemistry (LINOZ) and a cold tracer (Schröter et al., 2018; Braesicke and Pyle, 2003), which is activated when temperatures are below a threshold temperature of 195 K. The cold tracer indicates air masses whose conditions are conducive to polar stratospheric clouds, heterogeneous chlorine activation and thus chemical ozone depletion. The cold tracer is characterised by a lifetime of 2 days and declines exponentially when temperatures rise above the threshold temperature to account for chlorine deactivation. This way, the full chlorine chemistry on stratospheric clouds is imitated by using the simplified approach of the cold tracer, rather than explicitly calculated. Furthermore, a passive ozone tracer is simulated ("$O_3$ passive") that is only transported and not affected by chemistry.

For qualitative comparisons with clouds observed by GLORIA, the sum of specific cloud ice content ($q_i$) and snow mixing ratio ($q_s$) is used to generate a cloud mask (Table 1). Furthermore, we compare the ICON-ART variables specific humidity ($q_v$), passive specific humidity tracer ("$H_2O$ passive"), ozone tracer ("$O_3$ tracer"), and passive ozone ("$O_3$ passive") with the corresponding GLORIA data (Table 2). Since $q_v$, "$H_2O$ passive", $q_i$ and $q_s$ are reinitialised at 0 UTC, the model data shown in the direct comparisons with GLORIA represent short-term forecasts with lead times of ~12 to 21 hours (depending on point in time during flight) that are interpolated to the corresponding geolocations of the GLORIA observations along the flight track. In contrast, the "$O_3$ tracer" and "$O_3$ passive" data are simulated continuously and integrate the effects of transport, mixing, and chemical processes (the latter for "$O_3$ tracer" only).

## 2.3 EMAC chemistry-climate simulations

The ECHAM/MESSy Atmospheric Chemistry (EMAC) model is a numerical chemistry and climate simulation system that includes submodels describing tropospheric and middle atmospheric processes and their interaction with oceans, land and human influences (Jöckel et al., 2010). It uses the second version of the Modular Earth Submodel System (MESSy2) to link multi-institutional computer codes. The core atmospheric model is the 5th generation European Centre Hamburg general circulation model (ECHAM5, Roeckner et al., 2006). In this study we used EMAC (ECHAM5 version 5.3.02, MESSy version 2.52, see Jöckel et al., 2010) with T42L90MA and T106L90MA resolution, i.e. with a spherical truncation of T42 (corresponding to a quadratic Gaussian grid of 2.8 by 2.8 degrees in latitude and longitude) and T106 (1.125 by 1.125 degrees) with 90 vertical hybrid pressure levels up to 0.01 hPa (approx. 80 km). A schematic representation of the horizontal model grid is shown in Fig. 3. To simulate realistic synoptic conditions, surface pressure and various prognostic variables

(temperature, vorticity, and divergence) are "nudged" towards the ECMWF ERA-Interim reanalysis (Dee et al., 2011) above the boundary layer and below 1 hPa using a Newtonian relaxation technique.

The applied model setup includes a comprehensive chemistry scheme with gas-phase and heterogeneous reactions on Polar Stratospheric Clouds (PSCs) and comprises about 35 submodels, including the chemistry submodel MECCA (Sander et al.,
2011); the photolysis submodel JVAL (Sander et al., 2014); the submodel MSBM, mainly responsible for the simulation of PSCs (Kirner et al., 2011); the submodel CLOUD, based on the ECHAM5 cloud scheme, simulating large scale clouds (Roeckner et al., 2006); the submodel CONVECT, calculating the convection and convective clouds (Tost et al., 2006b); and the submodel SCAV, responsible for scavenging and wet deposition of trace gases and aerosols (Tost et al., 2006a).

We performed three different simulations from 1 July 2015 to 1 April 2016 (initialised with an older EMAC simulation which
was started in 1994 and perpetuated to recent years), thus including the Arctic winter 2015/2016 and the PGS campaign. In the first simulation (our "standard" simulation), we use the horizontal resolution of T106 (EMAC-STD). Additionally, we

**Table 1.** Data sets and cloud parameters (cirrus/ice clouds).

| Dataset | Cloud parameter | Unit |
|---|---|---|
| GLORIA | Cloud index | - |
| EMAC | Large scale cloud snow/ice content (iwc) + convective cloud snow/ice content (cv_iwc) | kg / kg |
| ICON-ART | Specific cloud ice content ($q_i$) + snow mixing ratio ($q_s$) | kg / kg |

**Table 2.** Data sets, trace gas products and sensitivity simulations.

| Dataset | Water vapour | Ozone | Nitric acid |
|---|---|---|---|
| GLORIA | $H_2O$ | $O_3$ | $HNO_3$ |
| EMAC-STD | $H_2O$<br>$H_2O$ passive[1] | $O_3$<br>$O_3$ passive[1] | $HNO_3$<br>$HNO_3$ passive[1] |
| EMAC-T42<br>EMAC-NOSCAV | $H_2O$ | $O_3$ | $HNO_3$ |
| ICON-ART | Specific humidity[2] ($q_v$)<br>$H_2O$ passive[3] | $O_3$ tracer<br>$O_3$ passive[1] | - |
| **Unit:** | ppmv | ppmv | ppbv |

[1] no chemical sinks and sources, no cloud microphysics
[2] reinitialised daily at 00 UTC using ECMWF IFS, no chemical sinks and sources
[3] reinitialised daily at 00 UTC using ECMWF IFS, no chemical sinks and sources, no cloud microphysics

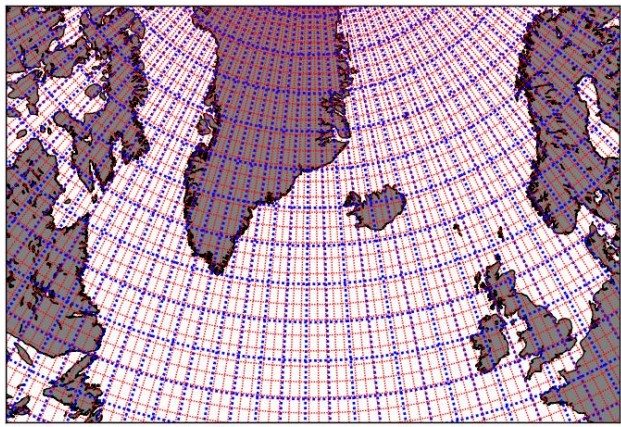

**Figure 3.** The EMAC standard and sensitivity simulations employed Eulerian grids with 106 (red) and 42 (blue) spectral coefficients. The T106 (T42) grid corresponds to a horizontal resolution of 125 km (310 km) at the equator. Due to the grid convergence, the zonal grid spacing is reduced towards the poles and amounts to ~40 km (~110 km) at 70° N.

performed two sensitivity simulations: First we reduced the horizontal resolution to T42 (EMAC-T42). In the second, we switched off the scavenging processes on ice particles, using again the T106 resolution (EMAC-NOSCAV). For comparisons with clouds observed by GLORIA, the combination of EMAC large scale cloud snow/ice content (iwc) and convective cloud snow/ice content (cv_iwc) is used (see Table 1). With respect to trace gases, the following EMAC variables are used: water vapour ($H_2O$), ozone ($O_3$), and gas-phase nitric acid ($HNO_3$) (Table 2). Furthermore, corresponding passive tracers are simulated, neglecting chemical sinks/sources and cloud microphysics ("$H_2O$ passive", "$O_3$ passive", and "$HNO_3$ passive").

### 2.4 Diagnostics

The vertical profiles of clouds and trace gases are combined into time-height cross sections of these parameters along the HALO flight tracks. For direct comparisons of synoptic and mesoscale patterns with the models, the ICON-ART and EMAC fields of the respective parameters are interpolated to the tangent point geolocations of the GLORIA observations (Fig. 1) to yield the corresponding model cross sections. In the vertical cross sections of the GLORIA data products, PV contours from the corresponding ECMWF reanalysis are superimposed to indicate the dynamical tropopause. For the model cross sections, PV is interpolated from the respective model output.

To quantify biases in the modelled trace gas distributions, the GLORIA and the interpolated model data of the variable under consideration are correlated against each other. In this manner, discrepancies between model simulations and observations can be identified as systematic deviations of data point populations from the respective 1:1 line. For a vertical assignment, i.e. to identify which data points are associated to the UT or LMS, the data points in the correlations are colour-coded with the corresponding PV values of the models. Furthermore, binned data points are shown to allow a clear identification of biases in the amount of overlapping data points.

The vertical resolution of the GLORIA data used here is in the order of 500 m, depending on altitude and parameter (see Johansson et al., 2018a), and therefore comparable with the vertical resolution of the simulations by both models in the tropopause region. Therefore, the use of 1-D averaging kernels in the vertical domain, such as often used in context of vertical profiles retrieved from satellite limb observations (e.g. Microwave Limb Sounder (MLS)) that are characterized by notably coarser vertical resolution is not expected to improve the comparison significantly. The absence of relevant overall systematic biases in the GLORIA data used here is furthermore confirmed by in situ comparisons (see Johansson et al., 2018a).

Due to the limb viewing geometry, strong horizontal gradients along the line of sight of GLORIA (i.e. towards the right hand side of the flight track) can affect direct comparisons of vertical cross sections of atmospheric parameters derived from the GLORIA observations and interpolated from the models at the tangent points. This effect can be taken into account by interpolating the model data with the help of 2-D averaging kernels (Ungermann et al., 2011, their Sect. 3.2). As discussed by Woiwode et al. (2018) in a case study where the mesoscale fine structure of a tropopause fold was investigated, the application of 2-D averaging kernels improves the model comparison only moderately if the observations are aligned such that horizontal gradients in the trace gas fields along the line of sight are small (see their Appendix A).

Aided by meteorological forecasts, the flight analysed here was planned so that the GLORIA observations were mostly aligned in such a way. This can be seen by comparing Fig. 1b with Fig. 4b, for example during the backward leg to Kiruna, when the GLORIA limb views were aligned along the direction of moist filaments above Greenland. Therefore, the viewing geometry allowed us to resolve the fine structures of the narrow filaments discussed in Sect. 4.3 remarkably well. Due to the suitable alignment of the GLORIA observations during the discussed flight and since the application of 2-D averaging kernels is computationally demanding (particularly in case of the GLORIA high spectral resolution chemistry mode observations that employ a large number of spectral sampling points), 2-D averaging kernels are not applied here. Therefore, local discrepancies between the GLORIA and model cross sections due to remaining effects by horizontal gradients along the line of sight cannot be excluded.

However, when the complete ensemble of GLORIA and model data points is analysed, such remaining effects by horizontal gradients are expected to cancel out on average due to the large amount of data points. Therefore, we consider the estimation of model biases in Sect. 4.4 to be robust.

# 3 Flight overview and meteorological analysis

Due to low planetary wave activity the Arctic winter 2015/2016 was extraordinarily cold (relative to preceding decades), and a strong polar vortex formed during November and December 2015 (Matthias et al., 2016). Cold conditions prevailed until February 2016. Then, three minor stratospheric warmings led to slightly warmer conditions in the polar vortex, but temperatures remained below the nitric acid trihydrate (NAT) PSC existence temperature (~195 K) on synoptic scales. In early March, the Arctic winter ended with the final stratospheric warming of the season. By mid-March, the vortex was displaced far off the pole and split. The "offspring" vortices decayed rapidly, resulting in a full breakup of the vortex remnants by early April (Manney and Lawrence, 2016).

PGS 14 was performed on 26 February 2016 from Kiruna, northern Sweden. Takeoff of the HALO aircraft was at 11:19 UTC and landing time was at 20:59 (flight duration of 9:40 h). The HALO flight track (anti-clockwise) and the tangent points of the GLORIA limb observations are shown in Fig. 1b. After takeoff, HALO headed westward (GLORIA pointing northward), crossed the Atlantic and Greenland, and continued its flight towards Canada. Then, at waypoint A, it turned southward (GLORIA pointing westward). Finally, after waypoint B, HALO turned back eastward and headed back towards Scandinavia (GLORIA pointing southward).

Figure 4 shows the meteorological situation on the day before the flight at 12 UTC (left column) and for the flight day at 18 UTC (right column), i.e. during the eastward flight leg back to Kiruna. The colour-coded contour plots in the upper row show ICON-ART $q_v$ at 10 km together with ICON-ART potential temperature (white contours) to visualise the dynamical situation in the UT/LMS region. West of the flight track, dry air masses characterised by high potential temperatures exceeding 340 K to 350 K indicate a deeply subsided air mass of the late-stage polar vortex, which was probed by the GLORIA observations during and around the southward-heading leg. As discussed by Johansson et al. (2019), these air masses were located within the vortex according to the vortex criterion from Nash et al. (1996). Relatively dry high-latitude LMS air masses are found above Greenland, the Arctic sea, and northern Europe and were probed by GLORIA during the westward- and eastward-heading legs (i.e., prior to waypoint A and after B, respectively). These high-latitude LMS air masses are interspersed with moist filaments that are connected to moist upper tropospheric air masses in the south and evolved in connection with a Rossby wave breaking event (see section 4.3). A broad filament of moist air stretches across the British islands, Iceland and Greenland on 25 February 2016 and has partly dissipated by 26 February 2016. During the return leg to Kiruna, GLORIA pointed first

towards upper tropospheric air masses (i.e. high specific humidity >40 ppmv) and then into dissipating filaments above central Greenland.

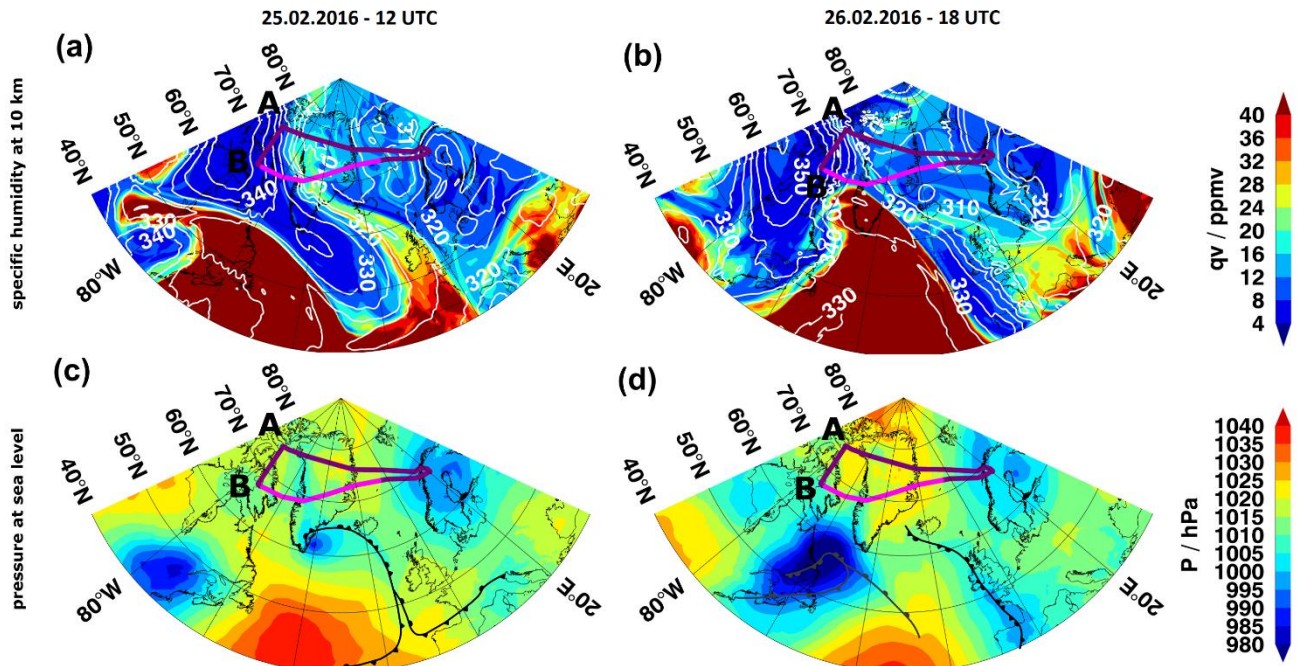

**Figure 4.** Meteorological conditions in the tropopause region and at sea level on 25 February 2016 (left column) and the flight day, 26 February 2016 (right column) as modelled by ICON-ART. Specific humidity is shown by colour contours and potential temperature is shown as white contour lines at 10 km altitude (a,b). Pressure at sea level is shown with selected warm fronts, cold fronts and occlusions (black and dark grey overlays) (c,d). The HALO flight track on flight day is indicated by a purple line. The section of the flight that covers the filaments observed by GLORIA and analysed in the model data in Sections 4.2 and 4.3 is highlighted in magenta in all panels.

The surface weather conditions are shown in the lower row of Fig. 4. On 25 February 2016, a well-defined low-pressure system is located above Scandinavia, and patchy weak high-pressure systems are found around central Greenland and Canada. A strong Azores high is located in the Atlantic Ocean together with a compact Icelandic low located at the southern tip of Greenland, accompanied by a notably positive North Atlantic Oscillation (NAO) index of +1.61 for February 2016. With a NAO value of +1.62 for the period from October to March, the winter 2015/16 ranks just within the top ten of the highest

seasonal values on record to date for this period of the year (both NAO values are retrieved from the record given at https://crudata.uea.ac.uk/cru/data/nao/values.htm; last access: 12 April 2021). An elongated occlusion stretches from South Greenland along Iceland to the Atlantic region near western Ireland. On the flight day, the front associated with the Icelandic low is fully occluded, while the situation above Greenland and Canada has only slightly changed. When comparing with the conditions in the UT/LMS region (Fig. 4, upper row), it can be seen clearly that the broad moist filament across Greenland on

25 February 2016 and its remnants on the flight day are connected to the occlusion associated with the Icelandic low. Above

the occlusion, moist tropospheric air masses are entrained into the surrounding LMS, and filaments of moist air are situated along the viewing direction of GLORIA during the return leg across Greenland.

Overall, at 10 km the air masses observed by GLORIA on 26 February 2016 comprise (i) the high-latitude LMS including patchy filaments, (ii) deeply subsided polar vortex air masses above Canada, (iii) upper tropospheric air masses above southern Greenland, (iv) moist air filaments above Greenland and associated with the occluded front of the Icelandic low, and (v) again high-latitude LMS air masses. Therefore, the GLORIA observations provide a unique opportunity to test the capability of ICON-ART and EMAC in simulating the Arctic winter UT/LMS region.

## 4 Observed and modelled cloud and trace gas distributions

### 4.1 Clouds

The vertical cross section of the GLORIA cloud index of the entire flight is shown in Fig. 5a. Cloud index (CI) values close to one are indicative of optically thick conditions, i.e. in the presence of clouds, whereas CI values approaching four and higher can be considered as cloud free conditions in spaceborne limb-sounding observations (Spang et al., 2004). In the case of airborne limb observations, CI values of 2 to 4 have been found to be suitable to separate between cloud-affected and cloud-free conditions in previous studies (Johansson et al., 2018 and references therein). In the case presented here, a cloud index of ~2.5 represents the threshold between cloud-affected and cloud-free conditions. High tropospheric clouds reaching the dynamical tropopause can be clearly identified around 12 UTC to 13 UTC, 14 UTC to 15 UTC, 16:30 to 17:30 UTC, and around 20 UTC, while a lower cloud system coinciding with a lower dynamical tropopause is detected directly at the beginning of the flight (prior to 12 UTC). A narrow band of low CI values is also visible around waypoint A around 8 km altitude. Further individual clouds are identified at lower altitudes between 17:30 and 19:30 UTC. Slightly reduced cloud index values at flight altitude (12 to 13 UTC and after 18 UTC) are the consequence of polar stratospheric clouds above flight altitude (Oelhaf et al., 2019) and are not indicative of cirrus clouds here.

In the following, we compare GLORIA cloud index values with cloud masks generated from the models in a qualitative way. The GLORIA cloud index is an optical quantity, while the model cloud masks are generated from the respective model outputs for condensed water in the solid state (see Table 1). Liquid water is not considered, since the temperatures in the focus region are well below the frost point, and there was no significant contribution of liquid water to the cloud masks used. A quantitative comparison (e.g. conversion of modelled cloud properties into spectral radiances and considering effects related to line-of-sight) is beyond the scope of our study that focuses on the ability of the models to reproduce the smaller scale structures.

We have set the threshold for the ICON-ART and EMAC model cloud mask at $10^{-9}$ kg/kg ice/snow water content (cloud parameters, see Table 1). On the one hand this is lower than the estimated sensitivity of $3\times10^{-6}$ g/m$^3$ for ice water content (IWC) in cirrus clouds of an IR limb sounder (Spang et al., 2015), corresponding to about $1\times10^{-8}$ kg/kg IWC at typical atmospheric conditions at 10 km altitude during the flight. Assuming that the representative concentration for a model grid-box volume is a mean of small-scale patches of enhanced concentrations, the choice of a small threshold value for the overall

volume seems sensible. On the other hand, it is higher than the lower in situ detection limit of cirrus clouds of $10^{-3}$ ppmv (Krämer et al., 2020; Schiller et al., 2008) corresponding to $6.2 \times 10^{-10}$ kg/kg.

The ICON-ART cloud mask represents the sum of cloud ice content ($q_i$) and cloud snow mixing ratio ($q_s$) interpolated to GLORIA geolocations along the flight track to 2-dimensional time-height cross sections (see Sect. 2.2 and Table 1). It shows the distribution of clouds of the ICON-ART simulation in a consecutive short-forecast mode along the flight track (see Sect. 2.2).

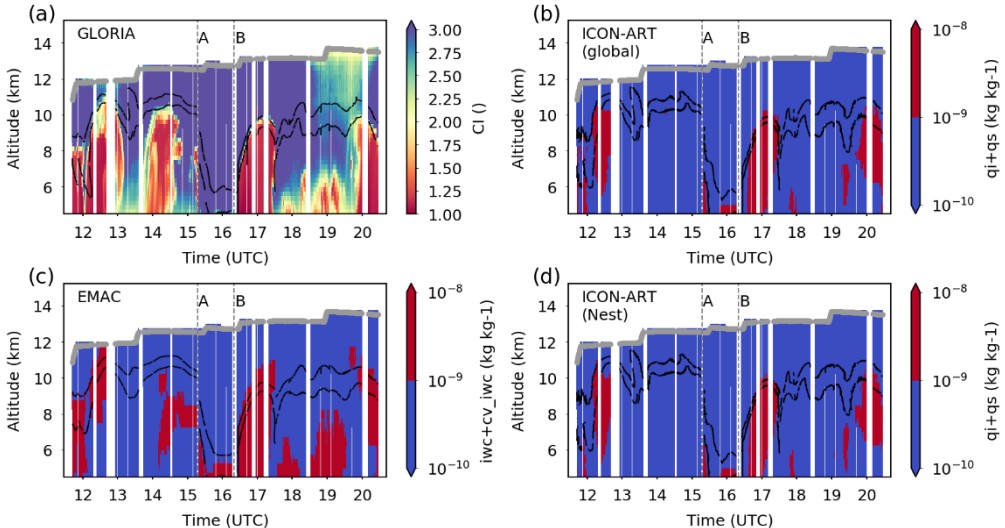

**Figure 5.** Qualitative comparison of clouds along flight track observed by GLORIA and cloud masks generated from ICON-ART and EMAC. (a) Vertical cross section of cloud index (CI) derived from GLORIA spectra. (b,d) Cloud mask constructed from global/nested ICON-ART domain of specific cloud ice content plus snow mixing ratio. (c) Cloud mask constructed from EMAC large scale snow/ice content plus convective cloud snow/ice content. Black lines: 2 PVU and 4 PVU isolines (lower and higher lines, respectively) from ECMWF reanalysis (a), ICON-ART (b,d) and EMAC (c) as indicators for the dynamical tropopause. Grey lines: HALO flight altitude.

In the global and nested ICON-ART domains (Fig. 5b and d), three of the four major cloud systems seen in the GLORIA observations can be identified, with differences in the vertical and horizontal extent. The results of the global and nested domains show only small differences, which are attributed to the simulations on the different grids and the interpolation from these grids with different widths.

The observed cloud system around 14 to 15 UTC below 10 km altitude is missing in both shown ICON-ART representations. Modelled cloud systems below approx. 10 km around 12 UTC to 13 UTC, 16:30 UTC to 17:30 UTC, and around 20 UTC agree well with GLORIA in the horizontal domain. Discrepancies in the large cloud system around 20 UTC below 6 km can be explained by the fact that no robust information on vertical cloud structure can be derived from GLORIA if optically dense cloud layers are located above. In such cases, lower limb views can be optically saturated, and low cloud index values may result although cloud-free conditions are present below. The same effect might explain differences between the observed and modelled cloud system between 12 and 13 UTC. We explain the fact that the vertically extended cloud system detected by

GLORIA around 14 UTC to 15 UTC is not reproduced by the ICON-ART simulation in both domains (global and nested) by a temporal mismatch in the simulated cloud systems (see Appendix A). Furthermore, the discrepancies might be explained partly by line-of-sight-related effects, since GLORIA accumulates light along extended limb views, while the model is interpolated at a certain geolocation. For the observed cloud systems at lower altitudes between 17:30 UTC and 19:30 UTC only weak indications are found in the ICON-ART simulation. Further high cloud systems prior to 12 UTC are barely reproduced in the ICON-ART simulation, and a simulated cloud at 16 UTC below 6 km is not confirmed by GLORIA.

The corresponding cloud mask of the EMAC standard simulation (EMAC-STD) with the T106L90MA resolution was generated by using the sum of the large scale cloud snow/ice content (iwc) and the convective cloud snow/ice content (cv_iwc) (see Table 1). As mentioned earlier the EMAC simulation uses a continuously nudged meteorology (see Sect. 2.3); however, the cloud variables are not nudged. As can be seen in Fig. 5c, the EMAC-STD reproduces the cloud patterns observed by GLORIA well. All of the observed cloud systems can be found in the cross section along the flight path generated from the EMAC simulation. Especially, the observed cloud system between 14 to 15 UTC, which is not reproduced by ICON-ART, is reproduced by EMAC, but with a different morphology and slightly displaced horizontally and vertically. Also, the lower clouds observed between 17:30 and 19:30 UTC are reproduced well by the EMAC simulation. As in the case of ICON-ART, a simulated low cloud system at 16 UTC is not confirmed by GLORIA.

In the EMAC simulation the modelled horizontal and vertical extents are mostly larger when compared to ICON-ART (e.g. prior to 12 UTC and 16:30 UTC to 19:30 UTC). The lower model resolution and lower time resolution of the output (1h for EMAC versus 0.25 h for ICON-ART) could be one possible explanation, making a positive cloud detection more likely (concerning the spatial coverage). Furthermore, the lower grid spacing is more comparable to the horizontal extensions of the GLORIA limb views, which results in a more consistent comparison in certain cases when cloud systems are located along the line-of-sight. The high cloud system prior to 12 UTC matches the GLORIA cloud index better than in the case of ICON-ART, while the cloud system in EMAC at 12 to 13 UTC appears higher than in the GLORIA and ICON-ART data, even exceeding the 2 PVU- and 4 PVU-isoline and reaching the GLORIA flight altitude. The clouds at 16:30 to 17:30 UTC and at 20 UTC also reach higher in the atmosphere in the EMAC cross section compared to the GLORIA and ICON-ART data, and again notably higher than the respective local dynamical tropopause. In these cases the ICON-ART cloud mask agrees better with the GLORIA observations.

Another proxy for the characterisation of detectable cloud systems in the model is looking at the cirrus cloud ice particle sedimentation events, which include the processes of nucleation, sedimentation and subsequent evaporation of cirrus cloud ice particles. As a consequence, local irreversible dehydration is found when ice particle growth removed water from the gas phase, and hydration is found at lower altitudes where the particles sublimate.

This is done in the following in the case of ICON-ART by using a passive water vapour tracer forecast in the constrained forecast mode as a reference. In addition, this analysis sheds light on the degree to which cirrus cloud ice particle sedimentation affects the modelled water vapour in the UT/LMS (cf. Sect. 4.2). The passive water vapour tracer does not account for cloud microphysics and therefore no nucleation, sedimentation and evaporation of hydrometeors. Residuals between the ICON-ART

specific humidity forecast (see Sect. 4.2) and the passive reference tracer show where microphysical processes altered UT/LMS humidity within the time frame of the forecast (i.e., the forecast lead time between ~12 h to ~20 h, depending on the flight section). It shows the cumulative effect of clouds and therefore indirectly the presence of cloud systems at the respective GLORIA geolocations during the time of the forecast on the day of PGS Flight 14.

Figure 6 shows the residual, i.e. the difference between ICON-ART (nested domain) specific humidity and the passive tracer

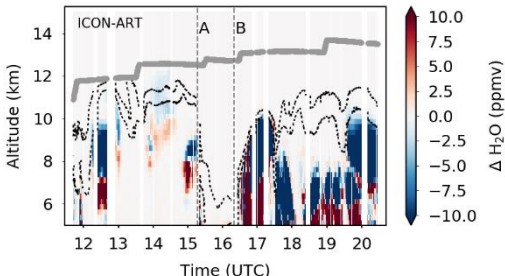

**Figure 6.** Modelled short-term changes in specific humidity due to cloud processes. Residuals between nested ICON-ART domain of specific humidity and corresponding H$_2$O tracer without cloud microphysics. Black dashed lines: ICON-ART 2 PVU and 4 PVU isolines (lower and higher lines, respectively) as indicators for the dynamical tropopause. Grey lines: HALO flight altitude.

without cloud microphysics. Negative residuals indicate regions which are depleted in water vapour due to cloud processes. Positive residuals show regions enriched in gas-phase water vapour due to sublimation of ice/snow particles. Negative and positive residual patterns clearly prove the generation and transformation of hydrometeors in the UT/LMS during the entire flight. Before the waypoint A, a strong pattern with residuals exceeding (~12 UTC to ~13 UTC and around 15 UTC) ±10 ppmv

is found, with weaker signatures in-between. After waypoint B, a sequence of distinct anomalies well exceeding ±10 ppmv is found until the end of the flight.

The comparison of Figure 6 with Figure 5a shows that this idealised ICON-ART diagnostic is a good proxy for the simulation of clouds in the model and does not require a threshold approach (as discussed above). However, it is an integrated quantity showing the history of "cloud events" on the respective day, whereas the cloud masks show "snapshots" of simulated

hydrometeors at the geolocations and time of the measurement. At a closer look all of the observed cloud systems coincide qualitatively with a corresponding cirrus cloud ice particle sedimentation pattern at the respective geolocations in the ICON-ART data. This means that there is evidence for the existence of all observed cloud systems in the ICON-ART simulation. However, as in the case of the ICON-ART cloud mask prior to 12 UTC, only weak indications of cloud systems are found here.

In particular, at 14 UTC to 15 UTC, where a cloud system detected by GLORIA is not reproduced by the cloud mask of ICON-ART (as described above and cf. Fig. 5a and 5b,d), barely resolved weak negative residuals reaching up to about -5 ppmv stretch even into the LMS and hint at drying of the uppermost troposphere and LMS by high altitude cirrus cloud ice particle sedimentation. Positive residuals of the same magnitude are found below between 9 km to 10 km, and another cirrus cloud ice particle sedimentation pattern in the direct vicinity at 14 UTC reaches further down to below 8 km. Therefore, these cumulative

patterns found in Fig. 6 support the idea that a cloud system has been present in the simulation at some time before the measurement during the day of PGS Flight 14.

There is also evidence in the ICON-ART data for the lower cloud system observed between 17:30 and 19:30 UTC (cf. Fig. 6 with Fig. 5a). Even though this cloud system is underestimated in the simulation (see Fig. 5b,d), Fig. 6 suggests that it has

been present at these locations at some time prior to the measurement in the simulation on the day of the flight.

The narrow cloud band at waypoint A, detected by GLORIA around 8 km, and also evident in the EMAC cross section (see Fig. 5c), is not visible in the ICON-ART cross section (cf. Fig 5a/c with 5b/d). However, again a strong signal of vertical redistribution of water vapour is visible in Fig. 6 at this geolocation, which again hints at the presence of this cloud system in the ICON-ART simulation at some time prior to the measurement. Thus, uncertainties in the timing of the ICON-ART forecast

might partly explain the discrepancies between GLORIA and ICON-ART here in addition to the other reasons discussed above. In the Appendix A we will further investigate this issue by sampling the models at the respective GLORIA geolocations with a negative time offset, to shed light on the history and development of the cloud systems in the models on the day of the flight and to prove that seemingly "missing clouds" in the ICON-ART data based on the cloud mask can be identified in the simulations just a few hours prior to the measurements.

Overall, the simulated cirrus cloud ice particle sedimentation patterns in Fig. 6 are consistent with the observed and modelled cloud systems in Fig. 5 and clearly show that modelled water vapour distributions in the UT/LMS are significantly modulated by more than $\pm 10$ ppmv in the UT and depleted by a few ppmv in the LMS. Therefore, cirrus cloud ice particle sedimentation clearly is a significant factor in modelled UT humidity on short-term time scales and also significantly affects LMS humidity. Cirrus clouds under cold conditions in the LMS have been found by many observations (e.g. Lelieveld et al., 1999; Kärcher

and Solomon, 1999; Spang et al., 2015) and are likely to affect LMS humidity by ice particle sedimentation (e.g. Kärcher, 2005). Furthermore, as discussed in the literature, convective hydration is known to affect the LMS and can drive air masses to saturation (Schoeberl et al., 2018; Zou et al., 2021). In summary, most of the major cloud systems observed by GLORIA can be identified qualitatively in both models. Remaining discrepancies between GLORIA and the models can be explained by horizontal and temporal mismatches of the cloud systems in the simulations and line-of-sight related effects of the GLORIA

observation. In particular the fact that ICON-ART fails to simulate the observed large cloud system at 14 to 15 UTC will be addressed in the Appendix A.

Note that line of sight related effects are capable of particularly strong influences on the comparison with respect to clouds. If, for instance, clouds were situated in front or behind the tangent point along the line of sight, this comparison would lead to a discrepancy between the model results and the measurements. Especially, complex small-scale cloud structures with strong

optical gradients (transparent/opaque) can differ in coverage and orientation when compared to the trace gas fields. Therefore, we consider the comparison of GLORIA cloud detection to the simulated clouds to be more difficult, in particular for small clouds or edges of clouds. Despite these limitations of the comparison, we mostly found good agreement between GLORIA and the models.

## 4.2 Trace gas distributions

In the following, we compare observations of water vapour, ozone and nitric acid with the respective simulated trace gases by ICON-ART and EMAC. For the former only water vapour, i.e. $q_v$, and ozone have been simulated.

Figure 7a-c show the water vapour, ozone and nitric acid distributions observed by GLORIA along the flight track. When compared with the cloud index plot (Fig. 5a), gaps in the retrieved trace gas distributions are explained by the fact that the presence of dense clouds precludes trace gas retrievals in the affected regions. Cloud filtering is applied here prior to the trace gas retrieval. Before waypoint A, moist tropospheric air masses extend to the dynamical tropopause, which is located mostly around 10 km in Fig. 7. Some moist "patches" are also found in the LMS here. In contrast, dry stratospheric air masses reaching down to ~6 km indicate a deeply subsided polar vortex remnant after waypoint A to slightly after waypoint B (cf. with Fig. 4). As discussed by Johansson et al. (2019) these air masses were situated within the vortex according to the vortex criterion from Nash et al., (1996). Afterwards, again a high tropopause around 10 km is found. The cloud system from 16:30 UTC to 17:30 UTC (cf. with Fig. 5a) is related to the moist tropospheric air masses above southwestern Greenland (cf. with Fig. 4). In the subsequent flight segment above central Greenland between 17:30 to 19:00 UTC, a highly textured LMS is found. Narrow moist filaments of tropospheric air reach as far as ~2 km into the LMS, and the dynamical tropopause altitude oscillates along the flight track. Afterwards, a more homogenous tropopause and water vapour distribution is found until the end of the flight.

The ozone distribution (Fig. 7b) shows a converse pattern compared to water vapour. At tropospheric altitudes, low ozone mixing ratios are found, while ozone mixing ratios above the tropopause increase with altitude. Also, in the ozone distribution, the deeply subsided polar vortex remnant from waypoint A to slightly after waypoint B can be clearly identified by high ozone mixing ratios reaching down towards ~6 km. From 17:30 UTC to ~19:00 UTC, filaments of low ozone correspond to the structures of enhanced water vapour (Fig. 7a) and reach even up to the flight altitude, therefore deeper into the LMS than the filaments seen in the water vapour distribution. For nitric acid (Fig. 7c), a similar pattern is found as for ozone, but with a higher contrast and more pronounced filaments from 17:00 UTC to ~19:00 UTC.

Furthermore, the nitric acid distribution shows a local maximum at and below flight altitude from ~15 UTC to ~17 UTC. As discussed by Ziereis et al. (2021), during this flight nitrified air masses were observed in situ at flight altitude during the first leg and the last leg, while descended denitrified air masses were observed at flight altitude during the central part of the flight. The local maximum in the GLORIA HNO$_3$ distribution below flight altitude between about 15 and 17 UTC is interpreted as subsided nitrified air masses that were located below denitrified air masses at flight altitude. As discussed by Braun et al. (2019) and Ziereis et al. (2021), both, nitrified and denitrified air masses were found in the LMS until March 2016.

As shown in Figure 7d, the overall distribution and mesoscale structures in the ICON-ART specific humidity forecast on the global R2B6 grid agree well with water vapour detected by GLORIA. Recall that $q_v$ was reinitialised using operational ECMWF IFS data at 00 UTC. The location of the strongest gradient in water vapour (roughly the transition from red to yellow shadings) is matched well during the entire flight. This applies also for subsided air masses from waypoint A to slightly after B. Therefore, the water vapour distribution suggests that the dynamical structure of the late-stage vortex air masses is modelled

in a realistic way by ICON-ART. During the return leg to Kiruna, excellent agreement is found for the narrow moist filaments and structures stretching into the LMS between 17:30 UTC and 19:00 UTC.

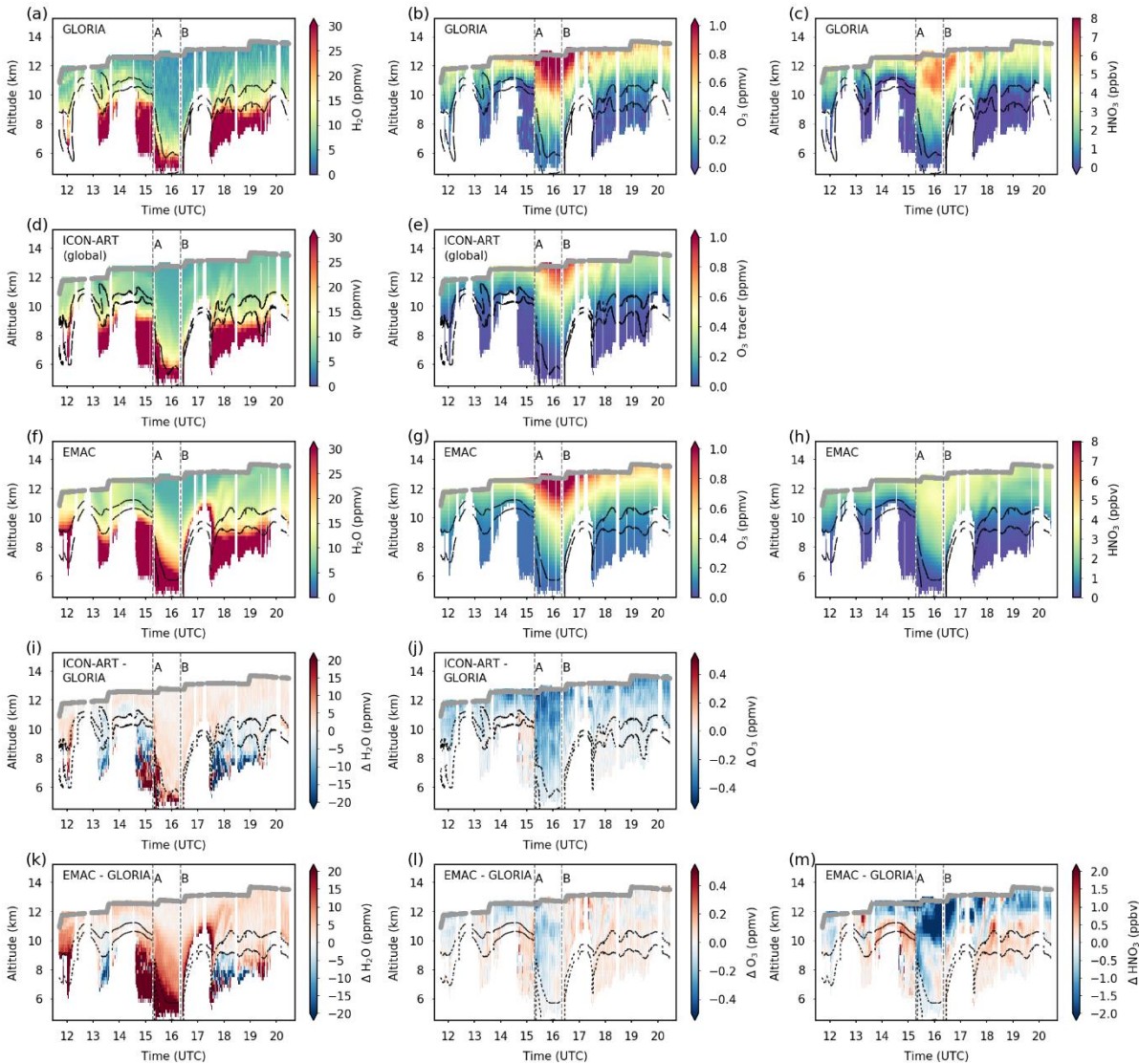

**Figure 7**. Observed and modelled trace gas distributions. GLORIA observations of water vapour, ozone and nitric acid (a-c). ICON-ART (global R2B6 grid) short-term forecast of specific humidity (d) and free-running simulation of ozone using simplified ozone depletion parameterisation (e). EMAC free-running simulations of water vapour, ozone and nitric acid (f-h). Residuals between the shown model data and GLORIA observations above (i-m). Black lines: 2 PVU and 4 PVU isolines (lower and higher lines, respectively) from ECMWF reanalysis (a-c), ICON-ART (d,e) and EMAC (f-h) as indicators for the dynamical tropopause. Grey lines: HALO flight altitude.

Keeping in mind that water vapour is simulated by EMAC continuously (i.e. no reinitialisation at 00 UTC and not nudged), the EMAC-STD simulation also reproduces the observed water vapour distribution well (Fig. 7f). Naturally, fewer details are

5   found in the EMAC simulation due to the lower horizontal resolution. The subsided air mass from A to slightly after B is

reproduced by EMAC. However, moister air masses with water vapour > 20 ppmv reach altitudes higher than those observed by 1-2 km. Furthermore, stratospheric air masses above the dynamical tropopause appear slightly moister in the EMAC simulation when compared to GLORIA and ICON-ART, and moist air masses reach above the dynamical tropopause in the vicinity of the cloud system around 17 UTC. Surprisingly, the moist filaments and structures seen in the GLORIA and ICON-ART data between 17:30 UTC and 19:00 UTC can be identified broadly in the EMAC simulation.

The continuous ICON-ART ozone simulation (i.e. no reinitialisation at 00 UTC) on the global R2B6 grid also matches the mesoscale patterns seen in the GLORIA observations (Fig. 7e), with however systematically lower volume mixing ratios. Again, the deeply subsided air masses from waypoint A to slightly after waypoint B can be clearly identified by higher ozone mixing ratios reaching down to lower altitudes. Similar filaments and structures as seen in the GLORIA observation between 17:30 UTC and 19:00 UTC are identified, with however fewer details and fine structures. The EMAC ozone distribution (Fig. 7g) matches the GLORIA observations well within the limitations of the model resolution, as already discussed by Johansson et al. (2019). Here, absolute mixing ratios agree quite well with the GLORIA observations. All major structures are reproduced, and weak indications are found again for the filaments and structures from 17:30 UTC to 19:00 UTC. The overall ozone mixing ratios in the EMAC simulation are higher when compared to ICON-ART and closer to the absolute values observed by GLORIA.

The nitric acid distribution simulated by EMAC (Fig. 7h) matches the overall structure seen in the GLORIA data only qualitatively. Systematically lower mixing ratios are found in the EMAC data, and local maxima seen in the GLORIA observations between 14 and 19 UTC are barely reproduced. This is probably due to the fact that EMAC underestimates nitrification of the LMS in this particular winter. A similar underestimation of nitric acid simulated by EMAC was found for the Arctic winters 2009/2010 and 2010/2011 as discussed in Khosrawi et al. (2018), and also in the comparison to GLORIA measurements of research flight 21 on 18 March 2016, described in Khosrawi et al. (2017). However, the observed narrow filaments with low nitric acid reaching into the LMS between 17:30 UTC and 19:00 UTC are again reproduced partly by the model.

Residuals between the model simulations and the GLORIA data are shown in Fig. 7i-m. A systematic moist bias is seen in ICON-ART in the LMS, while variable positive and negative residual patterns are found below the tropopause (Fig. 7i). Note that there is hardly any variation in the residual in the region of the narrow filaments above the tropopause from 17:30 to 18:30 UTC (Fig. 7i) due to the excellent agreement with GLORIA. ICON-ART ozone (Fig. 7j) shows a systematic low bias above the tropopause, while weak positive and negative residual patterns are found below the tropopause. EMAC $H_2O$ shows a predominantly positive bias at all altitudes except for subsections below the tropopause from 13 to 14 UTC and 17:30 to 19:00 UTC (Fig. 7k). From 17:30 to 18:30 UTC, noticeable residual patterns result above the tropopause since the filaments appear broader and with a slightly different shape when compared with GLORIA. For EMAC ozone, weak positive and negative residuals are found at all altitudes (Fig. 7l). EMAC $HNO_3$ is predominantly underestimated at altitudes higher than ~ 1 km above the 4 PVU level, while a systematic overestimation is found in the tropopause region below (Fig. 7m).

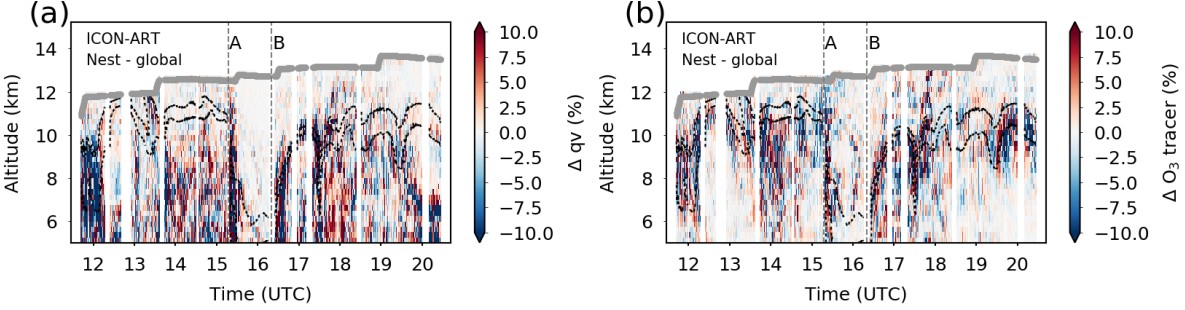

**Figure 8.** Residuals between the ICON-ART nested R2B7 and global R2B6 domains of $q_v$ (a) and $O_3$ (b). Black lines: 2 PVU and 4 PVU isolines as indicators for the dynamical tropopause. Grey lines: HALO flight altitude.

To investigate potential differences between the global R2B6 and the nested R2B7 ICON-ART domain, differences between these grids are depicted in Fig. 8. Mesoscale patterns in the residuals of $q_v$ (Fig. 8a) and $O_3$ (Fig. 8b) in the tropopause region and, in the case of $q_v$, in the regions where clouds were present (compare Fig. 5), are attributed to finer/coarser representation by the different model grids and the subsequent interpolation to the GLORIA geolocations. Overall, no significant systematic biases are identified.

In summary, the dynamical situation is represented well by both models (with either consecutive ICON-ART forecasts or continuously nudged EMAC simulations) within the limitations of their horizontal resolution. Both models clearly reproduce the observed strongly subsided air masses in the western part of the flight and the narrow filaments between 17:30 UTC and 19:00 UTC. Here, complementary patterns are found in the water vapour distribution when compared to ozone and nitric acid. Water vapour in the LMS is overestimated by EMAC, and ozone is underestimated by ICON-ART. Furthermore, EMAC clearly underestimates nitric acid and barely reproduces nitrification patterns seen in the GLORIA data.

### 4.3 Troposphere-to-stratosphere exchange region

Close-ups of the GLORIA, ICON-ART (nested R2B7 domain) and EMAC-STD trace gas distributions are presented in Fig. 9. In Figure 9a, two stronger moist filaments reaching into the LMS up to ~12 km are seen between 17:30 and 18:30 UTC, with a weaker filament in-between at ~18 UTC. The typical horizontal extent of the filaments along the flight direction is only 50-100 km just above the tropopause. During the further course of the flight warped regions of the dynamical tropopause are identified until ~19 UTC.

The ICON-ART simulation of specific humidity in the nested domain reproduces the vertical and horizontal extent as well as maximum mixing ratios very well (Fig. 9d). Even the weak filament in-between the more developed filaments can be clearly identified. However, overall water vapour mixing ratios are slightly higher when compared to GLORIA. In the EMAC simulation, the two major filaments can be weakly identified, and warping of the dynamical tropopause is weaker (Fig. 9f).

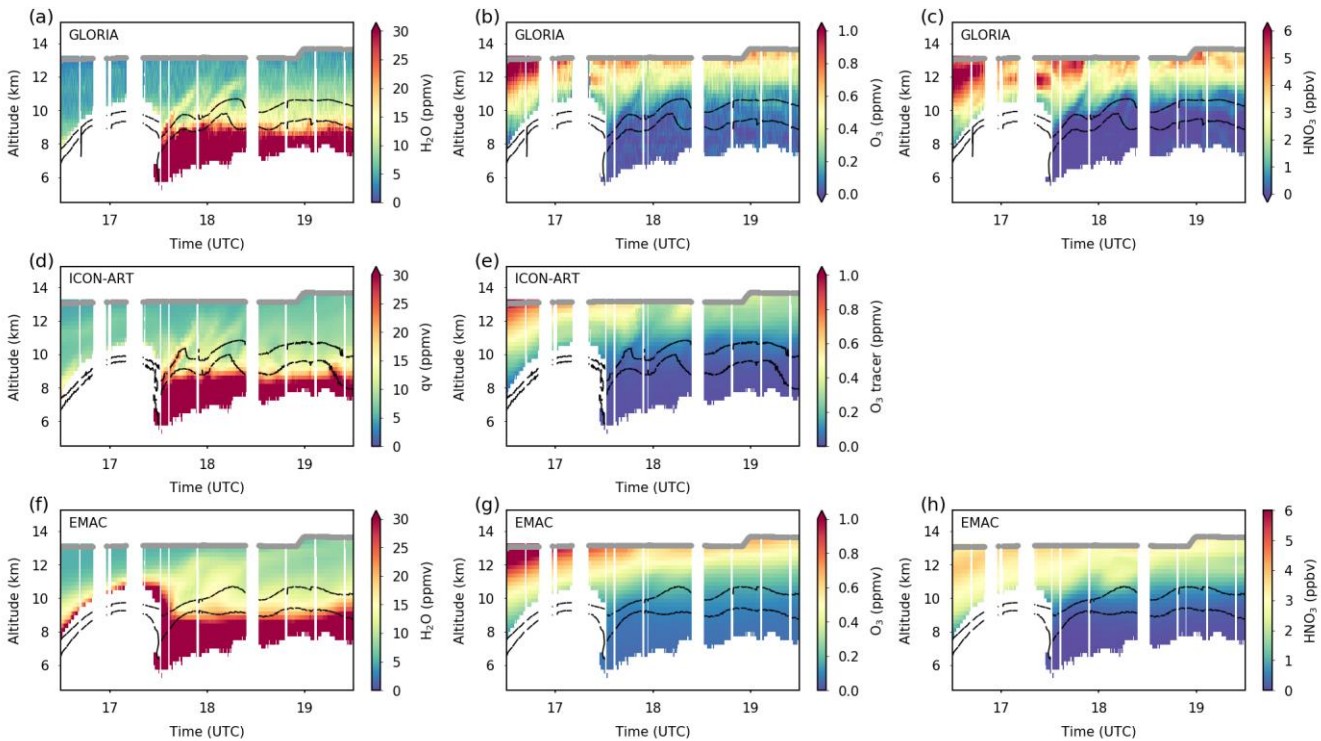

**Figure 9.** Close-ups of troposphere-to-stratosphere exchange region between 17:30 and 19:30 UTC. In the case of ICON-ART, the nested data is shown. For legend, see Fig. 7.

However, it has to be remembered that the horizontal resolution of the EMAC simulation is T106, which is lower than that of the ICON-ART R2B7 nest by about a factor of 5. Overall absolute water vapour mixing ratios are clearly overestimated by EMAC.

The GLORIA ozone distribution shows detailed fine structures close to the flight altitude. Structures low in ozone correspond to the high water vapour structures and extend further to flight altitude (Fig. 9b). The combination of ozone and water vapour data clearly shows that air masses characterised by tropospheric moisture levels reach deeply into the LMS and are connected to variations in the dynamical tropopause. Tropopause folds and steps in the tropopause are regions where isentropic levels cross the tropopause and jet streams. They are known bidirectional exchange regions between the tropopause and stratosphere (e.g. Shapiro, 1980; Keyser and Shapiro, 1986) and to contribute to transport and mixing of tropospheric air into the LMS such as diagnosed e.g. by Werner et al. (2010), Krause et al. (2018), and Jing et al. (2018) (note however that a net exchange from the LMS to the troposphere dominates).

The simulation of ozone in the nested ICON-ART domain reproduces the same sequence of filaments, with however lower mixing ratios and less fine structure. EMAC reproduces the filaments around 17:30 UTC to 18:30 UTC only faintly, while

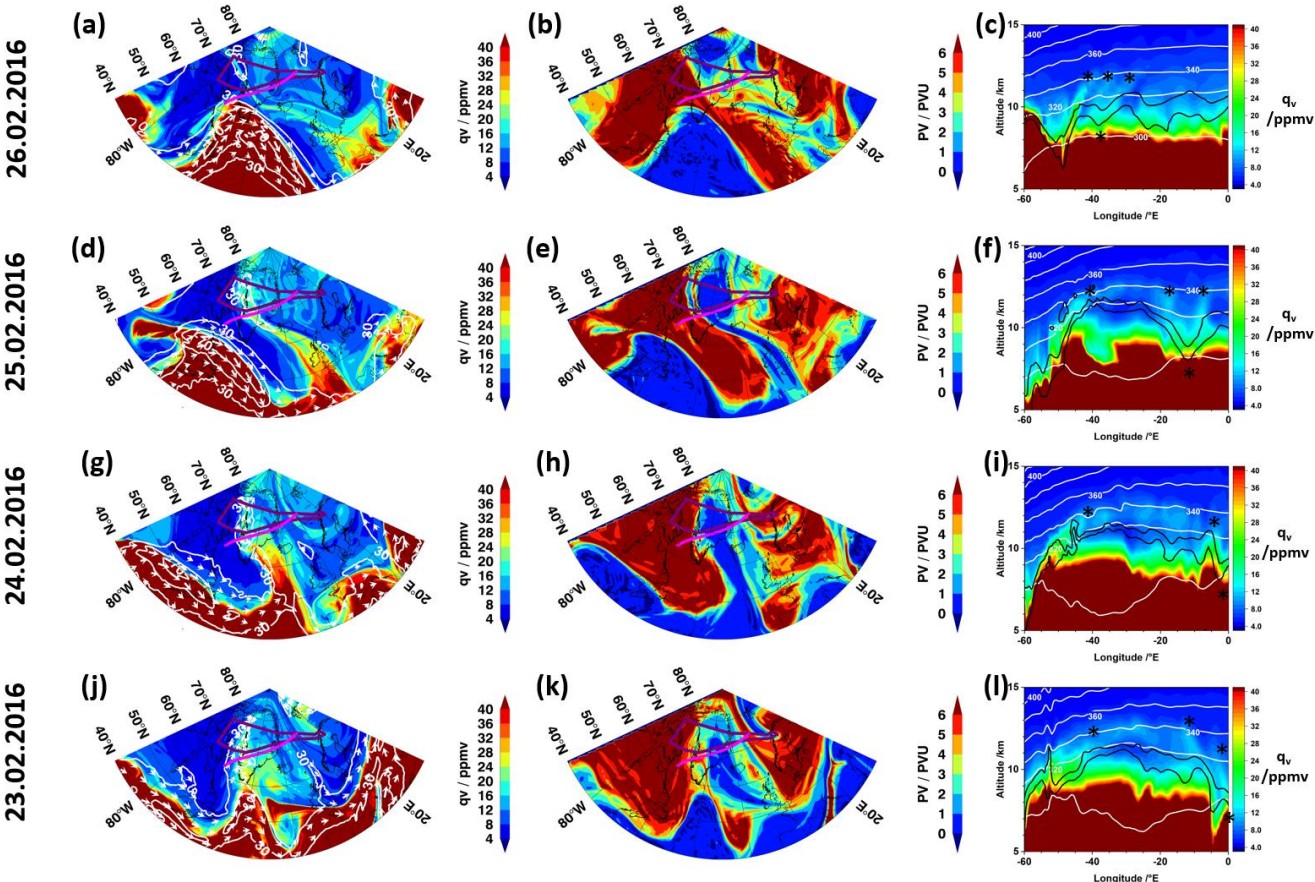

**Figure 10:** Evolution of filaments in nested ICON-ART domain. (a,d,g,j) Horizontal distribution of $q_v$ (coloured contour) and horizontal wind speed (white contour lines, in intervals of 20 m s$^{-1}$, and arrows) and (b,e,h,k) PV (coloured contour) at 10 km altitude. (c,f,i,l) Vertical distribution of $q_v$ (colored contour, in ppmv), potential temperature (white contour lines, in intervals of 20 K), and 2 and 4 PVU isoline (lower and upper black line) as indicator for the dynamical tropopause. Purple lines in in the left and middle column indicate the flight track and magenta lines the location of the vertical cross sections shown in the right column. Stars (c,f,i,l) indicate features in these panels which correspond with features in the other panels out of these. The model data is shown at 12 UTC of the dates indicated in the left.

observed absolute mixing ratios are matched well. Finally, the GLORIA close-up in Figure 9c shows a highly structured nitric acid distribution. EMAC again broadly captures the filaments, while mixing ratios are clearly underestimated and local maxima are barely reproduced (Fig. 9h).

In summary, Figure 9 shows that ICON-ART using the R2B7 (~20 km resolution) nest is able to resolve mesoscale fine
5  structures with a horizontal extent of less than 100 km. In the case of specific humidity, a similar degree of detail is achieved when compared to GLORIA, while fewer details are found in the simulation in the case of ozone. Given the lower resolution of the nudged T106 simulation of the EMAC model, we find that this model also reproduces dynamical structures at the lower

edge of its resolution. Clear evidence for structures resulting from troposphere-to-stratosphere exchange is found. Deviations in the trace gas distributions from both models are found and are quantified in the following section.

The evolution of the filaments seen in the GLORIA and model data is analysed with the help of ICON-ART. Figures 10a,d,g,j show the horizontal distribution of water vapour and horizontal wind from 23 until 26 February 2016 at 10 km altitude. The

wind contours south of ~60°N show the polar jet with meridional undulations, characteristic of a midlatitude Rossby wave (e.g. Gabriel and Peters, 2008; Wirth et al., 2018), which also manifests in the gradients of $q_v$ and PV (Fig. 10b,e,h,k). It separates moist upper tropospheric air masses in the south (high $q_v$, low PV) from dry stratospheric air masses in the north (low $q_v$, high PV). On 23 February 2016, the water vapour distribution in a ridge above southern Greenland is patchy, the jet is split into a northern and southern branch, with the northern branch carrying moist tropospheric air northward (Fig. 10j). The

ridge formed previously in a complex Rossby wave pattern above North America (not shown). The evolving moist filament is elongated towards the pole in the following two days (Fig. 10g,d). At the same time, the moist upper tropospheric air masses in the south move on eastwards, while an occlusion forms at the Icelandic low at south-eastern tip of Greenland in front of the ridge connected with the Azores high (see Fig. 4c). The wind speeds of the resulting northward-moving jet stream band in Fig. 10a decrease, resulting in the narrow moist filaments found at the flight day above central Greenland and a weak jet stream

band in the northwest. Moist upper tropospheric air masses associated with the ridge above south of Greenland on 23 February 2016 (Fig. 10j) and the moist filament (Fig. 10g,d) are framed by strong PV gradients (compare Fig. 10k,h,e). Only a narrow filament with weak PV gradients remains at the flight day (compare Fig. 10a with Fig. 10b).

In the region of the moist upper tropospheric air masses south of Greenland and the evolving broad filament with low PV towards the pole on the following days (Fig. 10k,h,e,b), the PV distribution shows meridional overturning of the PV gradient

that frames the moist upper tropospheric air masses. The pattern suggests poleward breaking of a cyclonically sheared Rossby wave (e.g. Gabriel and Peters, 2008 and references therein). Thereby, a separate isolated large patch of low PV values above west Greenland and the Atlantic on 23 February 2016 (Fig. 10k) combines with the moist upper tropospheric air masses with low PV in the south and seems to result from another Rossby wave breaking event that had previously occurred. As a consequence, a long broad filament with low PV stretches up to 80°N on the following days (Fig. 10h,e). On the flight day, a

patch of low PV north of Greenland has been cut off almost completely from the moist upper tropospheric air masses in the south (Fig. 10b).

The vertical cross sections shown in Fig. 10l,i,f,c correspond with the magenta lines in the left and middle column. The locations of the cross sections were chosen with the intention to cover the area sampled by GLORIA and to capture the connected atmospheric structures in the vicinity that are discussed above. As can be seen from the vertical cross sections shown

in Fig. 10l,i,f,c, the evolving filaments are framed in the west and east by steep gradients in tropopause height. The larger moist filament originates from the region around the jet stream band that branched away during the Rossby wave breaking event (compare Fig. 10j,g,d,a). It is aligned nearly parallel to the 320 and 340 K isentropic levels on 23 February 2016 (Fig. 10l). At lower altitudes, the 300 K isentropic level crosses the dynamical tropopause in the west in Fig. 10l,i,f,c. As discussed by Shapiro (1980), such regions provide suitable conditions for bidirectional cross-tropopause exchange. At higher altitudes, the

PVU isoline crosses the 320 K isentropic level in the same region and suggests conditions suitable for isentropic transport across horizontal PV gradients also here.

Local oscillations of the isentropic levels on 23 February 2016 between 55 and 50°W are attributed to a mountain wave above southern Greenland (Fig. 10l). During the following days, the moist filament aligns steeper across the isentropic levels (Fig. 10i,f). In the same region, oscillations of the dynamical tropopause become weaker on 24 February 2016, and patches of enhanced PV remain until 25 February 2016. On 26 February 2016, the remaining narrow moist filament is aligned along a newly formed tropopause fold in the west and reaches steeply into the LMS (Fig. 10c). Note however that the air masses seen in these panels are also modulated by horizontal transport in meridional direction and therefore have to be interpreted in combination with the maps shown in the left and middle row of Fig. 10.

The other two filaments on 23 February 2016 in the east are associated with a tropopause fold remnant in the east (Fig. 10l). The tropopause fold remnant declines during the subsequent days, moves west (Fig. 10i,f) and joins with the newly formed tropopause fold in the west on 26 February 2016 (Fig. 10c). Since these two filaments are aligned steeply across the isentropic levels already on 23 February 2016, they are interpreted as older structures that were previously formed in a similar way like the stronger filament in the west.

Overall, the vertical cross sections in Fig. 10l,i,f, c show that the filaments observed by GLORIA evolved along steep gradients of the dynamical tropopause in connection with Rossby wave breaking. The larger filament in the west evolved during a Rossby wave breaking event, where moist air tropospheric masses were transported horizontally into the Arctic LMS along the jet stream under conditions suitable for cross-tropopause exchange. The other two filaments are interpreted as older structures in connection with a tropopause fold remnant in the east that probably evolved during a previous Rossby wave breaking event.

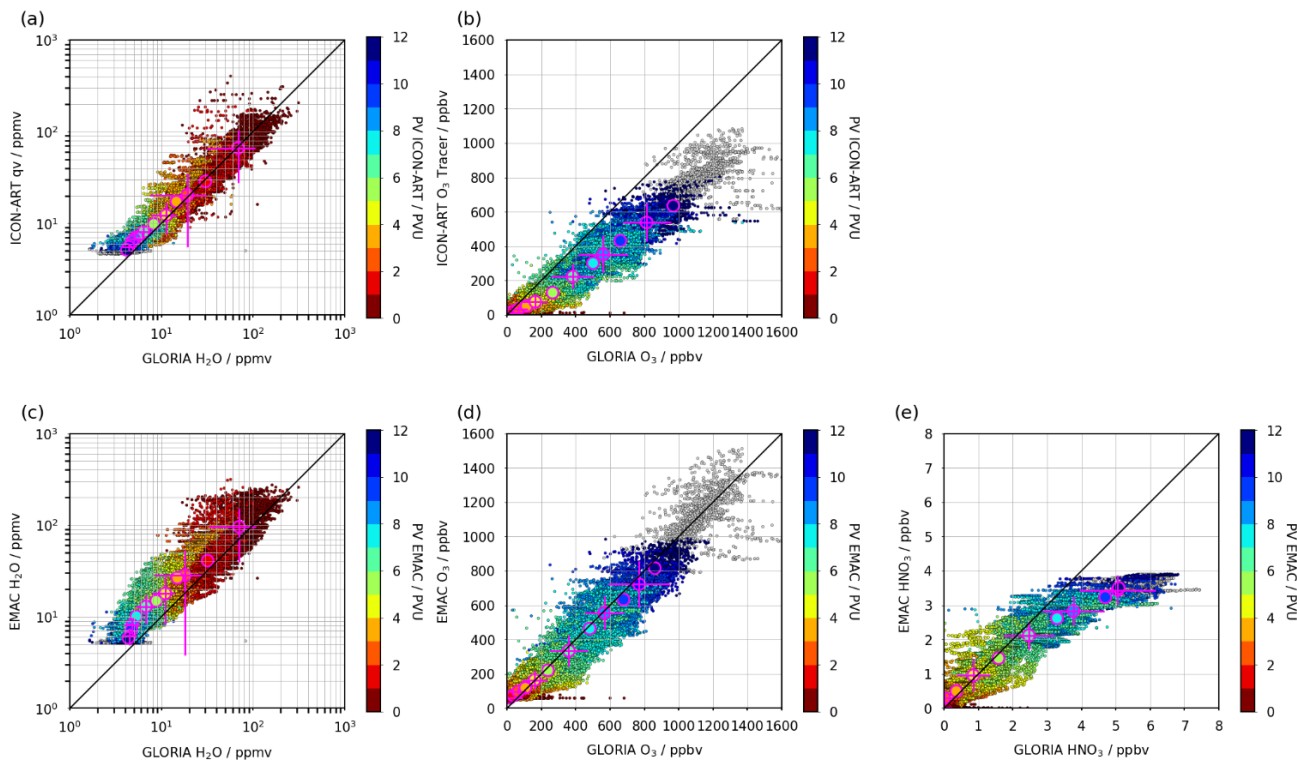

**Figure 11.** Correlation of GLORIA H₂O, O₃ and HNO₃ to corresponding ICON-ART and EMAC output variables. The large data points framed in magenta are a binned representation of the small data points. Magenta bars indicate the standard deviation of the binned data points. Colour-coding: PV from corresponding model.

## 4.4 Quantification of model discrepancies and sensitivity studies

By scattering and correlating modelled mixing ratios with the observed values, model discrepancies (and likely biases) can be quantified as deviations from the ideal 1:1 line (Fig. 11). Furthermore, a dynamical context in the vertical domain is provided by colour-coding the data points with corresponding PV values.

For ICON-ART specific humidity, excellent agreement is found for high tropospheric water vapour levels (Fig. 11a). At PV levels higher than ~4 PVU, a systematic moist bias is evident in the ICON-ART model data. The systematic offset at the high PV levels is attributed to the same systematic moist bias that is known to affect the ECMWF and other weather forecast systems (e.g. Stenke et al., 2008). It is not unexpected that this bias is translated into the ICON-ART simulation, since the simulation is done in a constrained forecast mode reinitialised from ECMWF IFS data. The correlation of EMAC $H_2O$ with GLORIA water vapour (Fig. 11c) shows a systematic moist bias in the model from the troposphere (low PV values, red) up to the highest stratospheric air masses accessed (high PV values, blue). The LMS moist bias is higher than in the case of the ICON-ART forecast (Fig. 11a). Towards the highest PV levels accessed, both model data sets move somewhat towards the ideal 1:1 line. For ozone in the nested ICON-ART domain (Fig. 11b), a systematic low bias is found that increases with PV. This is attributed

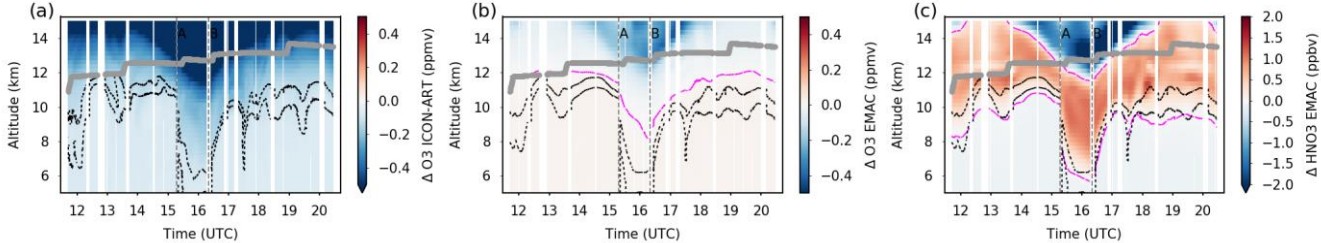

**Figure 12**. Modelled ozone depletion and changes in nitric acid due to chemical and microphysical processes. Residuals between (a) ICON-ART $O_3$ tracer and passive $O_3$ tracer, (b) EMAC $O_3$ and $O_3$ passive, and (c) EMAC $HNO_3$ and $HNO_3$ passive. Black dashed lines: ICON-ART/EMAC 2 PVU and 4 PVU isolines (lower and higher lines, respectively) as indicators for the dynamical tropopause. Grey lines: HALO flight altitude. Magenta lines indicate the zero contour.

to the simplified ozone depletion parameterisation. For the T106 EMAC simulation the agreement in ozone with GLORIA measurements is very good (Fig. 11d). Here, the data points are well scattered around the 1:1 line at all PV levels. Small groups of data points with larger deviations at high PV values are attributed to fine structures in the LMS, which are seen in the GLORIA data, but which are not resolved by the model (cf. e.g. Fig. 9b versus 9g).

To quantify the simulated cumulative impact of ozone depletion and nitrification of the LMS in the ICON-ART and EMAC simulations during the entire winter until the flight date, corresponding passive tracers are simulated (Fig. 12). Residuals between the "active" tracers (i.e. chemical and microphysical processes activated) and the corresponding passive tracers (only dynamical processes act on them) indicate the cumulative net changes due to the processes considered in the "active" case.

In the ICON-ART simulation, the "active" ozone tracer simulation shows systematically lower mixing ratios than the "passive" ozone tracer (Fig. 12a) at all altitudes due to modelled ozone depletion. Above the dynamical tropopause, the difference increases from -0.1 ppmv to more than -0.4 ppmv and shows that the ozone deficit increases vertically within the late-stage polar vortex.

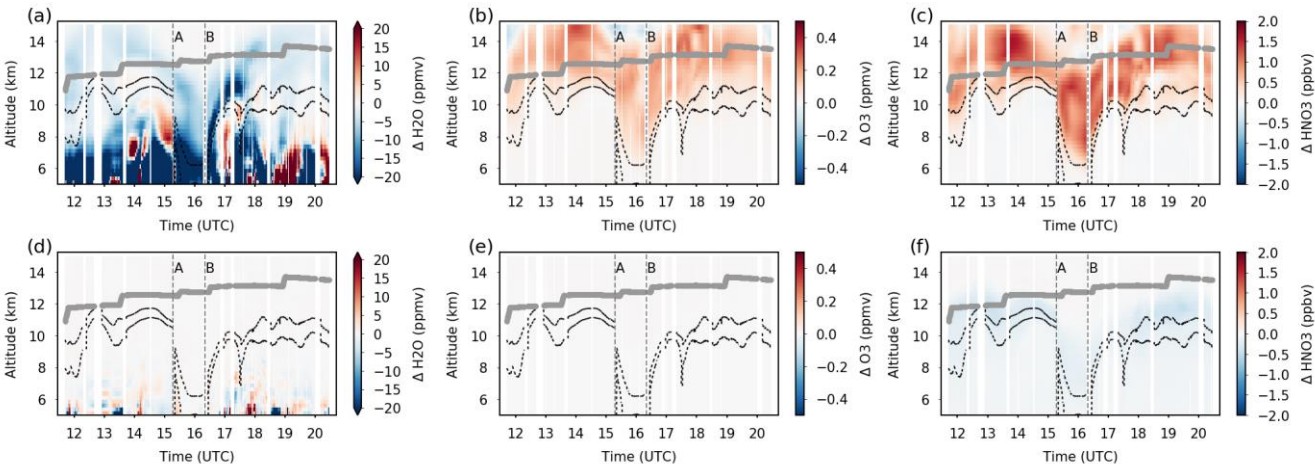

**Figure 13.** Modelled differences in $H_2O$, $O_3$ and $HNO_3$ due to lower resolution (a-c, T106 minus T42 resolution) and neglecting scavenging processes in clouds (d-f, EMAC-STD minus EMAC-NOSCAV). Black dashed lines: EMAC 2 PVU and 4 PVU isolines (lower and higher lines, respectively) as indicators for the dynamical tropopause. Grey lines: HALO flight altitude.

In the EMAC simulation (Fig. 12b), the residual is close to zero in the troposphere, in the tropopause region and also at lower levels of the LMS. Only in the deeply subsided vortex remnant around waypoint A and B is ozone significantly lower in the "active" simulation, which is indicated by residuals exceeding -0.2 ppmv. The fact that ICON-ART residuals are more negative in the LMS than in the case of EMAC and recalling that overall EMAC ozone agrees well with GLORIA (see Fig. 9d) suggests that the simplified ozone scheme used by ICON-ART overestimates ozone depletion in the LMS.

While EMAC nitric acid agrees well with GLORIA in the troposphere, a systematic low bias is found above the troposphere that strongly increases with altitude (Fig. 11e). The bias amounts to ~50 % at the highest PV levels of ~10 PVU under consideration and suggests that the observed nitrification of the LMS is not well reproduced. A similar bias has been identified by Khosrawi et al. (2017) in a comparison of EMAC with GLORIA results (PGS Flight 21).

The EMAC nitric acid residual shown in Fig. 12c clearly shows that this species is enhanced in the simulation by ~ 1 to 1.5

ppbv in large parts of the LMS as a consequence of nitrification by evaporated $HNO_3$-containing particles that sedimented from polar stratospheric clouds. In the middle of the flight, air masses that are affected by $HNO_3$ depletion reach into the LMS and are the consequence of subsidence of denitrified air masses in the polar vortex (see Khosrawi et al., 2017 and Ziereis et al., 2021). Above flight altitude, sequestration of $HNO_3$ on existing PSC particles might still play a role under sufficiently cold conditions.

Finally, the EMAC sensitivity simulations presented in Fig. 13 show that changing the model resolution from T106 to T42 exacerbates the LMS moist bias in the water vapour distribution (Fig 13a, compare Stenke et al., 2008) and results in

significantly lower mixing ratios in the LMS ozone (Fig 13b) and nitric acid distributions (Fig 13c) in the T42 simulation. A similar behaviour of EMAC was found in the stratosphere by Khosrawi et al. (2017), who stated that the T106 simulation agrees slightly better with Aura/Microwave Limb Sounder observations for both species.

Scavenging processes by cirrus cloud ice particles are capable of removing trace gases from the gas phase. Sedimentation of the ice particles is capable of removing the trapped gases from affected altitudes. While previous studies focused mainly on scavenging on liquid cloud droplets (Tost et al., 2010; Wang et al., 2010; Pierce et al., 2015; Kaiser et al., 2019), Tost et al. (2010), however, found $HNO_3$ values in the Northern hemisphere upper troposphere to be low due to uptake on ice particles and subsequent sedimentation. Thereby, relative changes were found to be large due to low absolute values there. In addition, the vertical redistribution of $HNO_3$ could induce secondary effects on other trace gases via chemical processes. In particular, altering $HNO_3$ could lead to changes in the budget of reactive nitrogen oxides ($NO_x$), which, in turn, could impact ozone (e.g. Kelly et al., 1991; Krämer et al., 2008; Schiller et al., 2008). Here, our goal is to test whether the effect of scavenging over ice on the trace gas composition is significant in the LMS in the EMAC simulation.

As can be seen in Figure 13d-f, simulated scavenging processes result in noticeable changes in the LMS only in the case of nitric acid. $HNO_3$ mixing ratios in a band of ~±1 km around the 4 PVU isoline are slightly lower by up to about 0.5 ppbv than in the standard simulation. Recalling that EMAC simulates here absolute mixing ratios of ~ 2 ppbv, this suggests that nitric acid is significantly higher in the LMS if scavenging processes by clouds are neglected. Even though $HNO_3$ in EMAC is underestimated in most parts of the LMS (see Sect. 4.2 and above), it is overestimated in most of the region between the 2 PV- and 4 PV-isoline and ~1 km beneath (see Fig. 7m). This, in turn, means that $HNO_3$ mixing ratios in the EMAC simulation are closer to GLORIA measurements in this region if scavenging processes are considered, and it hints that trapping of $HNO_3$ by high altitude cirrus clouds could play a significant role in the lower LMS. This would be consistent with the fact that cirrus clouds are known to occur also in the LMS (e.g. Lelieveld et al., 1999; Kärcher and Solomon, 1999; Spang et al., 2015, Zhou et al., 2021) and thus $HNO_3$ trapping is likely to take place here. Furthermore, LMS composition is known to be affected by troposphere-to-stratosphere exchange, which is likely to involve air masses that were previously affected by $HNO_3$ trapping in cirrus clouds, thereby resulting in less $HNO_3$ in the LMS when compared to a scenario without $HNO_3$ trapping. Our results are consistent with the results by Tost et al. (2010), who found a similar effect in the upper troposphere.

### 4.5 Suggestions for model improvement

In the following, the diagnosed model biases and suggestions for model improvement are summarised:

- ICON-ART $q_v$: Here, the water vapour is a short-term forecast based on ECMWF IFS data, and the moist bias found in the ICON-ART data is comparable with the same bias in ECMWF data. Therefore, no specific improvement for ICON-ART can be suggested here. Suggestions to improve the ECMWF data are provided in the literature (e.g., Dyroff et al., 2015; Woiwode et al., 2020).

- ICON-ART $O_3$: The ozone is modelled by the LINOZ-scheme, which represents a linearised ozone chemistry, and by using a cold tracer. The observed bias might be reduced by tuning this scheme. An optimized setup may be achieved by adaptation of the main parameters threshold temperature and lifetime of the cold tracer such that agreement with observations is improved (e.g. satellite observations such as MLS or field observations with suitable coverage).

- EMAC $H_2O$: The water vapour is simulated continuously in the EMAC model, i.e. it is neither reinitialised at 0 UTC nor nudged. The moist bias found in the EMAC simulation ranging from the troposphere to the LMS suggests that the cumulative impact of drying events in the entire altitude region is underrepresented in late winter. Such drying events might be precipitation events, which are dominated by ice and snow at the latitude and season associated with our case study. The parameterisation of ice nucleation and growth of ice particles might be optimised and tuned to improve the agreement with observations (e.g. satellite observations such as MLS or field observations with suitable coverage). Since our results show that the UT/LMS water vapour distribution is affected by model resolution in case of EMAC, a resolution-dependent tuning might be required.

- EMAC $O_3$: Ozone in the EMAC model agrees well with the GLORIA data. Therefore, no significant suggestion for improvement can be provided here.

- EMAC $HNO_3$: Nitric acid is systematically underestimated by the EMAC model in most parts of the LMS, while it is overestimated in the tropopause region and slightly above. The clearly noticeable negative bias of EMAC $HNO_3$ in the LMS suggests that downward transport of this species by sedimentation of NAT particles originating from polar stratospheric clouds (PSCs) with associated nitrification of the LMS is underrepresented. While considerable progress has been made in the representation of NAT in model simulations in recent years, significant uncertainties remain in the microphysical parameterisation of NAT particles in PSCs (Tritscher et al., 2021 and references therein). More field observations of NAT containing PSCs would be helpful to improve model physics including, among other factors, NAT nucleation rates, particle sedimentation characteristics and particle size distributions, and thereby simulate the associated nitrification of the LMS more realistically.

   The positive bias of $HNO_3$ in the tropopause region is even larger in EMAC-NOSCAV compared to EMAC-STD, i.e. results of EMAC-STD including scavenging processes are closer to the GLORIA observations in these regions. This suggests that scavenging processes of $HNO_3$ by high altitude cirrus clouds are relevant and might be underestimated in EMAC. An optimisation of the parameterisation of the scavenging process in the model with the help of observations might reduce this deficiency. Thereby, it should be taken into account that an optimisation of the representation of denitrification/nitrification by NAT particles might modulate the $HNO_3$ distribution here, too.

We propose to consider the model biases and deficits found here and our respective suggestions for future model development. As this work represents a case study, our findings hint at model deficiencies that might also be present in different seasons or

latitudes. Further observations and model validation studies are needed to investigate these issues and to pinpoint these deficiencies to the respective deficits in the parameterisations.

## 5 Discussion and conclusions

Using GLORIA observations taken during the HALO long-range flight on 26 February 2016, we test the ability of the atmospheric chemistry model ICON-ART and the CCM EMAC to model mesoscale dynamical features, the chemical composition and cirrus clouds and their impacts in the UT/LMS. The flight constitutes a multifaceted test case, covering deeply subsided air masses of the aged 2015/16 Arctic vortex, high-latitude LMS air masses, a highly textured region affected by troposphere-to-stratosphere exchange, and high-altitude cirrus clouds.

In both models, even though very different in their character, the dynamical situation, in particular, with the strongly subsided air masses in the western part of the flight, is simulated well. Here, the observed stratospheric air masses, characterised by low water vapour, high ozone and enhanced nitric acid mixing ratios, are reproduced.

The high-resolution ICON-ART setup (in a consecutive short-forecast mode) involving a R2B7 nest (approx. 20 km) reproduces mesoscale dynamical structures also quite well. Narrow moist filaments in the LMS observed by GLORIA at

tropopause gradients in context of a Rossby wave breaking event and in the vicinity of an occluded Icelandic low are clearly reproduced by the model. A more detailed analysis with ICON-ART shows that a larger filament in the west was transported horizontally into the Arctic LMS in connection with a jet stream split during poleward breaking of a cyclonically sheared Rossby wave. Further weaker filaments are associated with an older tropopause fold in the east. Given the lower resolution of the nudged T106 simulation of the EMAC model, we find that this model also reproduces these features at the limit of the

model resolution in a very reasonable way. All major cloud systems detected by GLORIA can be identified qualitatively in both models by cloud masks generated from the respective ice water content variables interpolated to the GLORIA geolocations. Remaining discrepancies between GLORIA and the models as well as between the two models are attributed to uncertainties in the modelled geolocations or timing of cloud scenarios as well as the limited qualitative comparison of the measured quantity cloud index with cloud masks generated from the models. We have demonstrated that residuals between

the active water vapour tracer and the corresponding tracer neglecting cloud microphysics in the ICON-ART simulation can be used as an alternative proxy for the presence of clouds, in terms of an integrated picture of the short forecast. In particular, this proxy hinted at a cloud system observed by GLORIA at 14 to 15 UTC, which is not present in the ICON-ART simulation at this particular time. However, a corresponding cloud system is found in the model data a few hours prior to the measurement at this particular geolocation. Both models tend to simulate cloud systems reaching higher above the tropopause than observed

by GLORIA and suggest that LMS humidity is significantly affected by cloud microphysics in the simulations. This is supported by the ICON-ART short-term sensitivity forecast neglecting cloud microphysics, which shows that LMS humidity can be depleted locally by cloud processes by 1-2 ppmv within less than 20 hours.

Overall magnitudes of UT/LMS humidity are reproduced well by the consecutive ICON-ART short-term forecasts (reinitialised at 00 UTC with ECMWF IFS) and the continuous simulations of EMAC water vapour. However, a systematic moist bias is found in the LMS in both models. The same moist bias is known for the ECMWF and other weather/atmospheric forecast systems and is a contributing factor to a cold bias there in medium-range forecasts with these systems (Stenke et al.,

2008). The fact that both models tend to simulate cirrus clouds reaching higher above the tropopause than observed by GLORIA might be related to the moist bias. Here, enhanced saturation versus the ice phase in the model simulations might be a reason for the cloud systems reaching to higher altitudes. Consistent with other studies (Roeckner et al., 2006; Polichtchouk et al., 2019), we find a higher moist bias in an EMAC simulation with a lower resolution (T42 instead of T106).

While the overall ozone mixing ratios of EMAC are in good agreement with GLORIA, the simplified ICON-ART $O_3$ scheme

LINOZ and the use of a cold tracer (Braesicke and Pyle, 2003) to imitate heterogeneous chemistry on PSCs systematically overestimate ozone depletion in the LMS by ~0.2 ppmv. This bias might be reduced by tuning of the LINOZ-scheme and/or the threshold temperature and lifetime of the cold tracer. Furthermore, EMAC nitric acid does not show nitrification of the LMS to the same extent as observed. This bias has already been documented in Khosrawi et al. (2017, 2018) by comparing EMAC to satellite data. The same problem has furthermore been found in a previous study for the same winter using the

CLaMS model (Braun et al., 2019), which suggests that microphysical properties of $HNO_3$-containing particles in polar stratospheric clouds resulting in denitrification of the stratosphere and nitrification of lower layers are not parameterised in a sufficiently realistic way.

We find that LMS composition modelled by EMAC is notably affected by model resolution. In addition to the enhanced moist bias, a reduction in horizontal resolution from T106 to T42 leads to a low bias in ozone, and an even more pronounced low

bias in nitric acid. This effect, concerning ozone and nitric acid, has been also found in Khosrawi et al. (2017), when compared to satellite data, with these authors however focusing on higher altitudes. These discrepancies might be overcome by resolution-dependent model tuning. Finally, our EMAC simulations show that neglecting scavenging processes by clouds has practically no impact on water vapour and ozone in the LMS, while nitric acid is noticeably depleted by ~0.5 ppbv if scavenging processes are activated in the simulation.

Overall, we find that ICON-ART and EMAC T106 are well suited for comparison to high resolution remote sensing aircraft data and are capable of simulating troposphere-stratosphere exchange in the context of Rossby wave breaking. Fine structures like the filaments seen in the GLORIA data between 17:30 and 18;30 UTC are reproduced well by ICON-ART and even modelled broadly by EMAC despite the much coarser resolution.

The GLORIA data were measured during a single flight on 26 February 2016 with a duration of 9 hours 40 minutes and a total

distance of ~8000 km. The flight covered a multifaceted scenario of the UT/LMS at high latitudes performed prior to the final major warming (Manney and Lawrence, 2016, and Matthias et al., 2016). Therefore, the presented comparisons of the GLORIA and model data can be considered representative for the polar UT/LMS at high latitudes in late winter prior to the vortex breakdown.

However, we find that accurate simulation of UT/LMS composition remains challenging and both models need to be further improved. We speculate that the reported biases and sensitivities might help to provide better forecasts and long-term projections by these and other models. The observed biases in ICON-ART $O_3$ and EMAC $H_2O$ might be reduced by improving the model physics and optimising parameterisations in combination with comparisons with observations (e.g. satellite observations such as MLS or field observations with suitable coverage). The EMAC simulation of $HNO_3$ might be improved by refining the microphysical representation of NAT with the help of further field observations. Continuing high resolution measurements of atmospheric trace gases and clouds are required to continue to test and further improve the models, so that they can be used for reliable projections of temperature trends in the UT/LMS and surface climate.

## Appendix A

In this section we want to get back to the comparison of observed clouds by GLORIA and modelled clouds by ICON-ART and EMAC. To prove that seemingly "missing cloud systems" in the ICON-ART model, in particular the cloud system at 14 to 15 UTC, had been present at some time prior to the measurement at the respective geolocations in the model, and to examine the evolution of clouds during the day of PGS Flight 14, we have sampled the model output of ICON-ART cloud variables ($q_i$ and $q_s$) and the EMAC cloud variables (iwc and cv_iwc) at the GLORIA geolocations, but with negative time offsets varying from -1 to -10 hours.

Figures A1 to A3 show the evolution of clouds in the ICON-ART (panels d-f) and EMAC (panels g-i) model at various times between -10 hours to -1 hours prior to the GLORIA measurements interpolated to the GLORIA geolocations, which are defined by altitude and time of measurement (in UTC) along the flight. For better comparison Figure 5, which corresponds to no time offset in the models, is again attached in Fig. A3.

The cloud system detected by GLORIA at 14 to 15 UTC corresponds to geolocations along the westward flight leg between central Greenland and approx. the west coast of northern Greenland (see Fig. 1), with GLORIA pointing to the north.

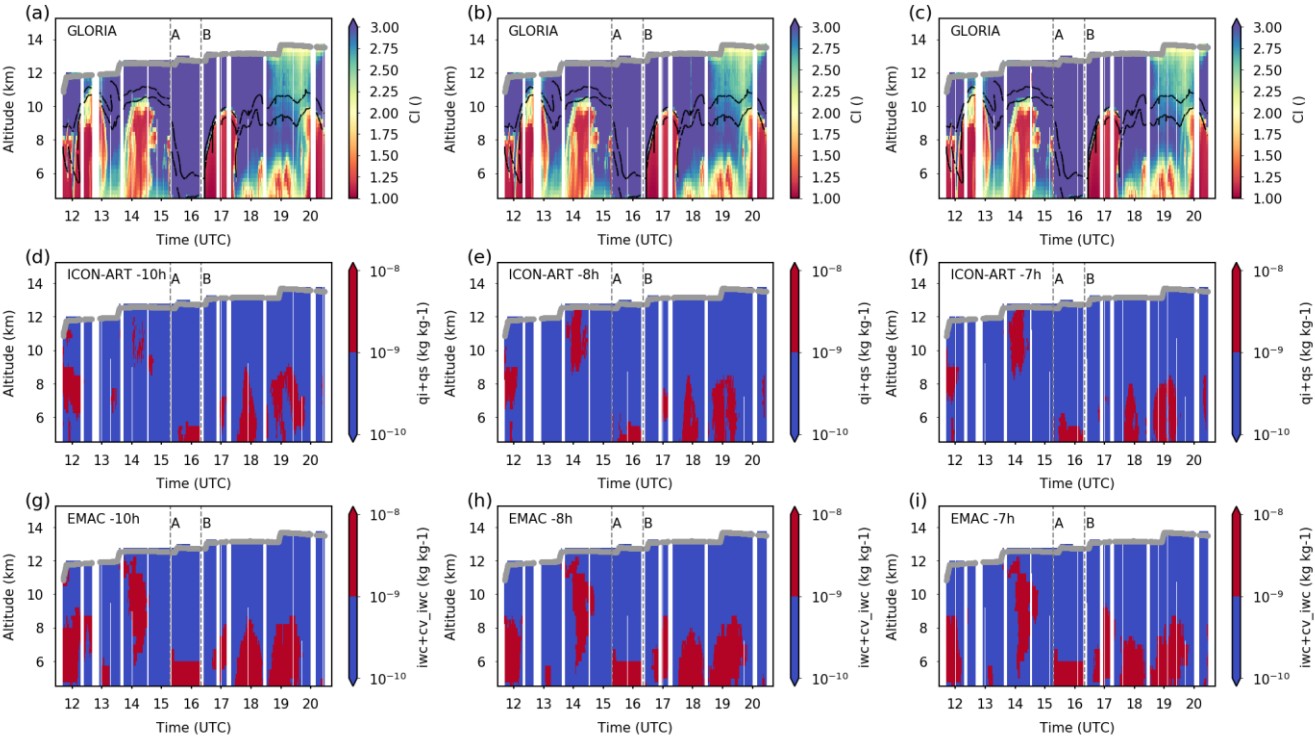

**Figure A 1.** Same as Figure 5, but model data (ICON-ART and EMAC) have been sampled with a constant time offset of -10, -8 and -7 hours during the interpolation to the GLORIA geolocations along the flight.

Inspection of the panels d-f in Figures A1 to A3 shows that a corresponding cloud system is forming about 10 hours before the measurement in the ICON-ART model and is growing until it reaches its maximum vertical and horizontal extent at about a time offset of -7 hours (Fig. A1f). It is also centered higher in the atmosphere than the cloud system measured by GLORIA. Afterwards (from -6 hours to -2 hours) the cloud system dissolves while subsiding into presumably warmer layers, until it completely vanishes at -1h hour (Fig. A3e). This proves that a corresponding cloud system is also present in the model data; however, it appears a few hours earlier at the particular geolocation. Discrepancies in the shape of the modelled cloud system from the observed pattern in the GLORIA cloud index might result partly from the optical thickness of the cloud top (which might offset the cloud index values below) and also the fact that the atmospheric scenario has changed in the considered time interval.

The corresponding cloud system in the EMAC simulation (Fig. A1-A3, panels g-i) appears with slightly different shape, but with remarkably larger vertical extent, reaching down deep into the troposphere to about 6 km altitude. A separate part appears close to flight altitude and seems to be connected to the main cloud system in the troposphere. The connected cloud system remains approximately constant from -10 to about -6 hours, where it breaks apart into two pieces (Fig. A2g). Afterwards, the upper part dissolves and vanishes at about -3 hours (Fig A2i), while the lower part subsides and decreases in shape to its tropospheric remnants at the time of the GLORIA measurements as depicted in Fig. 5.

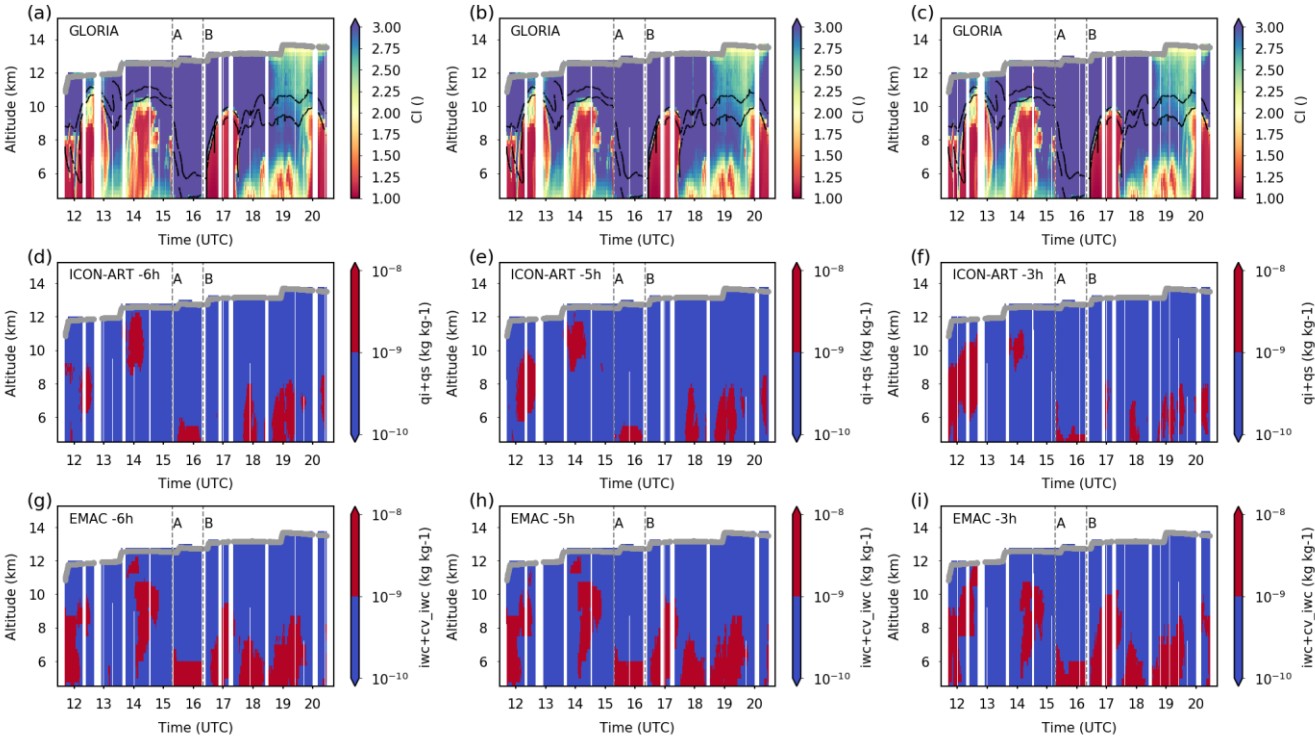

**Figure A 2.** Same as Figure 5, but model data (ICON-ART and EMAC) have been sampled with a constant time offset of -6, -5 and -3 hours during the interpolation to the GLORIA geolocations along the flight.

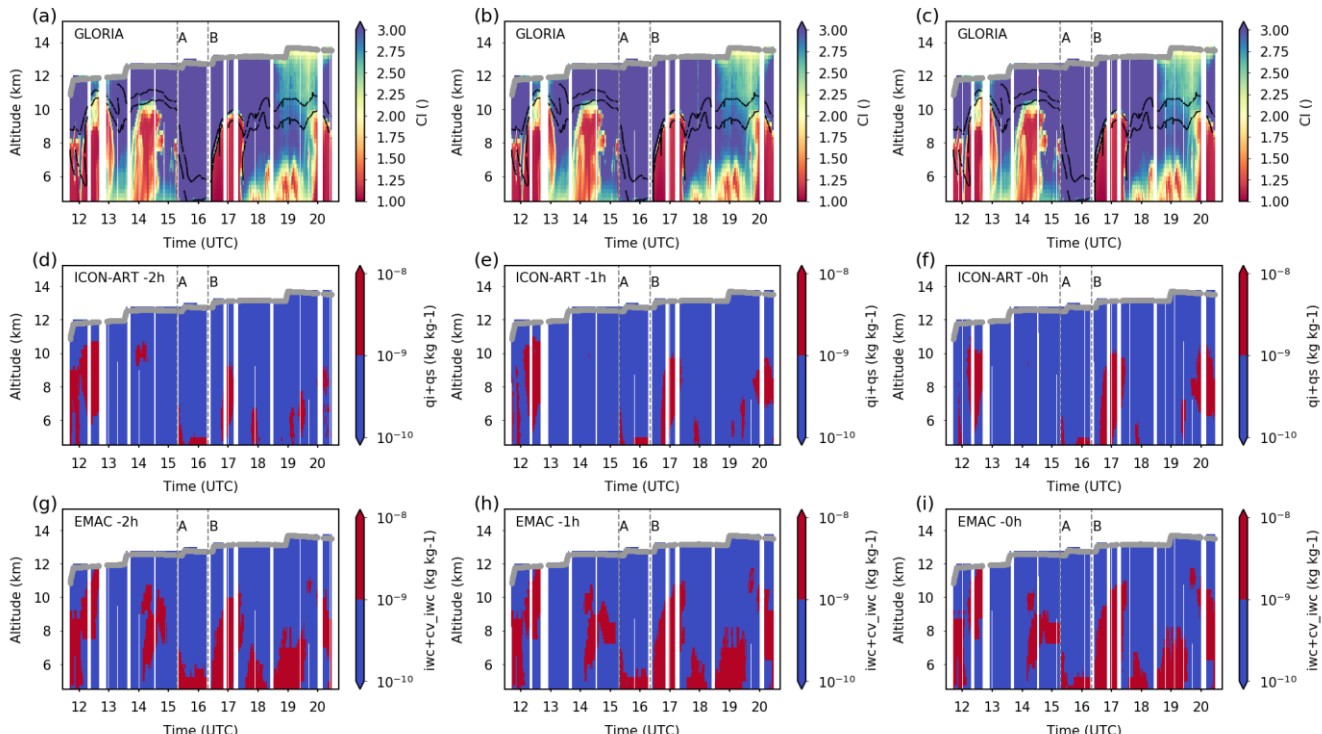

**Figure A 3.** Same as Figure 5, but model data (ICON-ART and EMAC) have been sampled with a constant time offset of -2, -1 and 0 hours during the interpolation to the GLORIA geolocations along the flight.

In Sect. 4.1 we also found hints that the lower cloud system between 17:30 and 19:30 UTC, which was underestimated in the ICON-ART cross section, is more pronounced at some time prior to the measurement. Inspection of Figures A1 to A3 yields that the corresponding cloud system has been more developed at these geolocations during the day of the flight, reaching its best resemblance to the GLORIA cloud index around -3 hours prior to the measurement.

However, we do not find any indications in Fig. A1 to A3 in the interpolated ICON-ART data (panels d-f) for a cloud located at waypoint A around 8 km altitude, which would be responsible for the large cirrus cloud ice particle sedimentation signal in Fig. 6, and which is also visible in the EMAC data (cf. with Fig. 5c).

In summary, this analysis yields that better resemblance of the ICON-ART cloud data to the GLORIA observations and EMAC simulations is found in some cases if model data of an earlier time step is considered.

In particular, the large cloud system observed by GLORIA at 14 to 15 UTC is reproduced in both the ICON-ART and EMAC model, however its vertical extent is much more pronounced in the EMAC model.

Both models show that this cloud system is subsiding with time, which is in accordance with the meteorological situation above Central Greenland (a high pressure system cf. Sect. 3).

*Data availability.* The data used here are available at the repository radar4KIT (https://doi.org/10.35097/454, Haenel et al., 2021). The GLORIA observations can also be accessed at the HALO database (https://doi.org/10.17616/R39Q0T, HALO consortium, 2016, last access: 16 April 2020) and at the KITopen repository (https://doi.org/10.5445/IR/1000086506, Johansson et al., 2018b, last access: 16 April 2020). The complete data of the ICON-ART and EMAC simulations are available on the Large Scale Data Facility (LSDF) of SCC. Access can be granted upon request.

*Author contributions.* FH and WW designed the study. FH and WW analysed the data with support from MH, PB, RR, SJ, AK, MW, OK, FK and JB. JB and MW performed the ICON-ART simulations. OK performed the EMAC simulations. HO, BMS and WW coordinated the HALO activities during PGS. FF-V and the GLORIA team performed the GLORIA measurements and operations. JU, AK and SJ performed the GLORIA level-1 data analysis. SJ performed the GLORIA chemistry mode trace gas retrievals (level-2 data) used here.  MH, AK, JU and SJ contributed to the discussion concerning the GLORIA cloud detection. FH and WW prepared the manuscript with contributions from the other authors.  FH, WW, MW and FK designed the figures. All authors helped with discussions and with finalising the manuscript.

*Competing interests.* The authors declare that they have no conflict of interest.

*Acknowledgements.* Atmospheric research with HALO is supported by the Priority Programme SPP 1294 of the Deutsche Forschungsgemeinschaft (DFG). The work of F. Haenel has been funded by the DFG project no. 316735585 (WO 2160/1-1) and by the project "Advanced Earth System Modelling Capacity" (ESM) with project no. ZT-0003 funded by the Helmholtz Association. We thank the GLORIA-Team and DLR-FX for performing the measurements and HALO flights during PGS. The EMAC and ICON-ART simulations were performed on the supercomputer ForHLR and with the help of the Large Scale Data Facility at the Karlsruhe Institute of Technology, both funded by the Ministry of Science, Research and the Arts Baden-Württemberg and by the Federal Ministry of Education and Research. The interpolations of EMAC and ICON-ART simulation data to the GLORIA tangent geolocations were performed on the bwUniCluster (2.0). The authors acknowledge support by the state of Baden-Württemberg through bwHPC. We thank ECMWF for providing the meteorological data used here.

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
