# Peer review of "Challenge of modelling GLORIA observations of UT/LMS trace gas and cloud distributions at high latitudes: a case study with state-ofthe-art models"

_Atmospheric Chemistry and Physics, 2021_

## Referee Comment (RC2)

**Review of "Challenge of modelling GLORIA observations of UT/LMS trace gas and cloud distributions at high latitudes: a case study with state-of the-art models" by Haenel et al.**

This manuscript uses GLORIA measurements taken on a PGS flight that sampled a diverse set of conditions in the UT/LMS to test the ability of two models, ICON-ART and EMAC, to simulate cloud structures and trace gas ($H_2O$, $O_3$, $HNO_3$) distributions. Both models are shown to reproduce the observations quite well; discrepancies between modelled and measured cloud and composition fields are quantified and their causes investigated. The paper is well organized and well written, and the figures are generally well prepared and support the discussion. I have only a few substantive issues that I would like to see addressed before the paper is accepted for publication; most of my comments are minor wording suggestions that will take very little time to act on.

Below both major substantive issues and minor points of clarification, wording suggestions, and grammar / typo corrections are listed together for each Section in sequential order.

Respectfully,
Michelle Santee

Abstract:
- p1L18-19: The wording "measurements taken in a challenging case study by the GLORIA" could be interpreted to imply that that flight was deliberately designed for this purpose, which I do not believe was the case. I think it would be better to simply say "measurements taken in a flight of the GLORIA" here and then add "challenging" in front of "multifaceted" in L23.
- p1L21: 2016, which --> 2016 that
- p2L3: moist-bias --> moist bias
- p2L7: changing of the --> changing the
- p2L8: play only a role in case of $HNO_3$ --> play a role only in the case of $HNO_3$
- p2L8-9: I agree that the representativeness of these results is an important question that should be explored. However, unless I missed it, this issue is not raised anywhere in the paper other than this sentence in the abstract. It should be acknowledged elsewhere as well, at least in the Discussion and Conclusions section if not somewhere in the main text of the paper.
- p2L10: projection --> projections
- p2L10: Although this study has certainly provided very useful information to characterize model biases, I am less convinced that it has really laid out sufficiently specific guidance to "define paths for further model improvements". See related final comment on the Conclusions below.

Section 1: Introduction:
- In a number of places in the presentation of background material (e.g., p2L13, p2L16, p2L21, p2L27, p2L28, p3L7), a few citations are given for very well-established concepts, but many other equally suitable papers could have been cited instead of or in addition to the ones listed. Obviously not all relevant papers can be referenced for these points, but "e.g." should be added in these lines to avoid giving readers the impression that the selected references are the only appropriate ones.

- p2L17-18: spread in these trends among models while perturbating ozone and other greenhouse gas abundances --> spread among modelled trends when ozone and other greenhouse gas abundances are perturbed
- p2L18: can be --> include
- p2L21: knowledge on --> knowledge of
- p2L23: compartment --> layer
- p2L24: On the winter --> In the winter
- p3L6: sedimentation … redistribute --> sedimentation … redistributes; "eventually" is not really needed here, but if the authors want to keep it, it should come before "changes"
- p3L12: I do know what is meant by "(in parts)" in this sentence; if I have understood the intent, this would be better as: "in part explicitly and in part by using" or "both explicitly and by using"
- p3L13: such models are the models ICON --> such models include ICON
- p3L34: I assume that the systematic biases referred to here are in the model fields (that is, the intention is not to use the model results to validate the GLORIA data), but that should be made explicitly clear, e.g.:  in the trace gas distributions --> in the modelled trace gas distributions

Section 2: Data and diagnostics:
- p4L10: the used model setups --> the model setups used
- p4L11: overview on --> overview of
- p4L15: aircrafts --> aircraft
- p4L24: combined to --> combined into
- p5L7: is operational --> has been operational
- p6L16: delete the comma after "winter"
- p6L19: air masses suitable for --> air masses whose conditions are conducive to
- p7L1: life time --> lifetime
- p7L9: does "between ~12 to 21 hours" mean "between ~12 and 21 UTC"?
- p7L16: atmosphere --> atmospheric
- p7L20: in the --> with; T106L90MA-resolution --> T106L90MA resolution
- Fig. 3 caption: The corresponding T106 (T42) grid corresponds --> The T106 (T42) grid corresponds; reduces --> is reduced
- p7L23-25: This is a very awkwardly worded and unclear sentence.  If I have understood it correctly, it would be clearer to say: "To simulate realistic synoptic conditions, surface pressure and various prognostic variables (temperature, vorticity, and divergence) are "nudged" towards the ECMWF ERA-Interim reanalysis (Dee et al., 2011) above the boundary layer and below 1 hPa using a Newtonian relaxation technique."  This formulation of the sentence also introduces the term "nudged", which is used later in the manuscript but is not currently defined.
- p7L26: a comprehensive chemistry --> a comprehensive chemistry scheme
- p7L27-30: Because of the complexity of the punctuation in these lines, some of the commas need to be replaced with semicolons (marked in red here): "… (Sander et al., 2011); the photolysis submodel JVAL (Sander et al., 2014); the submodel MSBM, mainly responsible for the simulation of PSCs (Kirner et al., 2011); the submodel CLOUD, based on the ECHAM5 cloud scheme, simulating large scale clouds (Roeckner et al., 2006); the submodel CONVECT, calculating convection and convective clouds (Tost et al., 2006b); and …".
- p9L9: combined to --> combined into

- p9L17: delete "that deviate"
- p9L18: e.g. --> i.e.

Section 3: Flight overview and meteorological analysis:
- p10L2: concerning the decades before --> relative to preceding decades
- p10L6: ended by --> ended with
- p10L9 and L11: take off --> takeoff
- p1011-12: headed westwards (GLORIA pointing to northward directions) --> headed westward (GLORIA pointing northward)
- p10L12-13: turned to a southward direction (GLORIA pointing to westward directions) --> turned southward (GLORIA pointing westward)
- p10L13-14: back to eastward directions and … southwards) --> back eastward and … southward)
- Fig. 4 and its caption:
  - It is very difficult to make out the overlaid white contours without greatly magnifying the plot – thicker lines and larger font for the labels would make them easier to see.
  - It would be convenient to have waypoints A and B marked on these maps as well as on Fig. 1.
  - is colour-coded in contour --> is shown by colour contours; delete "together"; occlusions --> occlusions (black overlays)
- p10L21: west- --> westward-; way-point --> waypoint
- p10L24: partly dissipates on --> has partly dissipated by
- p10L24-25: pointed subsequently towards … and into --> pointed first towards … and then into
- p10L29: the wording "going along with" is not completely clear.  Does this mean "consistent with" or "accompanied by"?
- p10L30-31: move "to date" to after "on record"
- p11L6: Why is the word "subsequently" used here?  The air masses observed during the flight contained these features – they were not observed subsequent to the flight.

Section 4: Observed and modelled cloud and trace gas distributions:
- p12L4-5: It is stated here that CI values "approaching four and higher" are indicative of cloud-free conditions.  Since the color scale in Fig. 5a saturates at CI=3.0, does that imply that on this flight GLORIA never encountered air masses that can be considered cloud-free?
- p12L9: It seems a bit odd to characterize air masses affected by clouds as having an "enhanced" CI since it is actually low values of CI that indicate the presence of clouds.
- p12L15: used cloud masks --> cloud masks used
- p12L18: threshold of the cloud mask for the ICON-ART- and EMAC-model at --> threshold for the ICON-ART and EMAC model cloud masks at
- p12L22: add a comma after "concentrations"
- Fig. 5b,c: I am a bit confused about why the tick marks on the cloud mask color bars are needed
- p12L33: Discrepancies between measured and modelled clouds at lower altitudes for the system around 20 UTC are attributed to GLORIA data being affected by optically dense cloud layers above, but couldn't this explanation be applied to the mismatch for other clouds as well, such as the one between 12 and 13 UTC?
- p13L1: delete the comma after "GLORIA"
- p13L3: delete the comma after "fact"

- p13L8: I'm not sure what "appear more sharply in the ICON-ART simulation" means, as the cloud systems in question barely register at all in the model cloud mask.
- p13L10: respective --> corresponding; EMAC-standard simulation (STD) --> EMAC standard simulation (EMAC-STD); T106L90MA-resolution --> T106L90MA resolution
- p14L1: Recalling that --> As mentioned earlier; 2.3), however the --> 2.3); however, the
- p14L2: the EMAC standard simulation (STD) --> the EMAC-STD
- p14L11: better comparable --> more comparable
- p14L19: delete comma after "model"
- p14L21: to which degree --> the degree to which
- p14L25: does "~12 h to ~20 h" mean "~12 UTC to ~20 UTC"? Also: accumulated --> cumulative
- p15L4-5: It is stated that "all of the observed cloud systems coincide qualitatively with a corresponding precipitation pattern at the respective geolocations in the ICON-ART-data", but the $\Delta H_2O$ diagnostic does not pick up the cloud system observed by GLORIA prior to 12 UTC.
- p15L8: Although I see weak negative residuals just below the tropopause, even with the figure greatly magnified it is difficult to discern non-negligible residuals above the tropopause.
- p15L10-11: vicinity is found at 14 UTC and reaches --> vicinity at 14 UTC reaches
- p15L11: support that --> support the idea that
- p15L18: again, hints --> again hints
- p15L20: beside --> in addition to
- p16L3-4: The use of the term "precipitation" is ambiguous here – I believe that the authors mean "cirrus cloud ice particle sedimentation", but that should be clarified. I also think that it would be appropriate to add discussion putting these results about "precipitation" affecting the humidity of the LMS into the context of previous studies that have examined the impact of convection and cirrus cloud processes on moistening / dehydrating the LMS (especially in light of my previous comment that I had trouble identifying these weak signatures in Fig. 6).
- p16L4: affects also significantly --> also significantly affects
- p16L5: the major cloud systems --> the major cloud systems observed by GLORIA
- p16L7: the ICON-ART lacks the simulation of the --> ICON-ART fails to simulate the
- p16L11: add a comma after "$q_v$"
- p16L12: It should be reiterated when Fig. 7 is introduced that the presence of optically thick clouds precludes trace gas retrievals, as comparison of the patterns in Figs. 5 and 7 shows.
- p16L13-14: The tropopause is located near 10 km in all panels of Fig. 7, not just 7a. In addition, use either "around" or "~", not both (see also L16).
- p16L17: south-western --> southwestern; part --> flight segment
- p16L18-19: reach by ~2km up into --> reach as far as ~2km into
- p16L21: complimentary --> complementary (but "converse" is probably a better word here)
- p16L25: reach up --> reach nearly up; add a comma after "altitude"
- p16L27-29: Although I don't doubt that some nitrification at lower altitudes occurred during this winter, the morphology of the $HNO_3$ distribution (Fig. 7c) does not seem very different from that of $O_3$ (Fig. 7b) to me, and abundances of both would be expected to be higher in the LMS than in the UT. Thus I am not certain what local maxima in $HNO_3$ are being referred to here. The specific signatures of nitrification in this figure should be clarified.
- p17L8: is the comparison ("higher") with respect to ICON-ART or GLORIA? Assuming the latter: reach here higher up by 1-2 km --> reach altitudes higher than those observed by 1-2 km

- p17L12: "schematically" is not quite the right word here – maybe "broadly" or "generally"?
- p18L3: fine-structures --> fine structures
- p18L19: complimentary --> complementary
- p18L31: Thereby --> However
- p18L32: which is by a factor of ~5 lower than that the ICON-ART R2B7 nest --> which is lower than that of the ICON-ART R2B7 nest by about a factor of 5
- p19L2: delete "respective"
- p19L3-4: It would be appropriate to acknowledge some of the previous studies that have also found substantial troposphere-to-stratosphere exchange associated with tropopause fold events; folded airmass structures reach --> airmasses in tropopause fold structures reach
- p19L6: shows highly --> shows a highly
- p19L7: I think that "broadly captures" or something like that would be better wording than "resolves in principle"
- p19L10: In case of --> In the case of
- p19L14: by both --> from both
- p20L5-8: I think that the flow would be improved by moving the introduction of Fig. 10 in these lines to after the end of the discussion of Fig. 9 on the following page. Also: on it) --> on them)
- p20L11: bias, which is known for the --> bias that is known to affect the
- Fig. 9 caption: EMAC and ICON-ART output --> ICON-ART and EMAC output
- p21L5: found and increases --> found that increases
- p21L8: fine-structures --> fine structures
- Fig. 10: It might be helpful to add an overlay outlining the zero contour, especially in Fig. 10b, since it is hard to tell where the EMAC ozone residuals change sign.
- p21L12: that ozone --> that the ozone
- p21L14: B, ozone is significantly --> B is ozone significantly
- p21L17: scheme by --> scheme used by
- p21L18-19: above the troposphere and strongly --> above the tropopause that strongly; amounts --> amounts to
- p21L21: while comparing --> in a comparison of
- p21L23: due --> due to; (de-) nitrification --> denitrification/nitrification
- p21L23-24: It is confusing to focus only on the evaporation of PSC particles here, as that leads to $HNO_3$ enhancement (renitrification). If I understand correctly, the modelled $HNO_3$ depletion associated with the subsided air mass encountered in the middle of the flight is being attributed to sequestration in existing PSC particles or permanent denitrification through their subsequent sedimentation. That should be clarified.
- Fig. 11 caption: T106 vs T42 resolution --> T106 minus T42 resolution
- p22L1-5: I am not convinced of the value of including the T106 vs T42 sensitivity test shown in Fig. 11a-c, as the benefit of using the higher resolution in EMAC has already been demonstrated in the Khosrawi et al. papers mentioned here. Why was it necessary to repeat this comparison?
- p22L2: "enhances" can have a positive connotation, hence: enhances --> exacerbates
- p22L4: stating --> who stated; MLS --> Microwave Limb Sounder
- p22L6: The findings about scavenging processes only being important for $HNO_3$ are presented here and later in Section 5 in a manner that suggests that they were unexpected. Did the

authors have any expectation that scavenging processes would affect the $O_3$ or $H_2O$ distributions? More background and context motivating this sensitivity test is needed.

- p22L7: ppbv in --> ppbv than in
- p22L8: Reminding --> Recalling
- p22L10: delete ", however,"; most parts of a region --> most of the region
- p22L11: delete "respective"; delete comma after "means"
- Fig. B1 and caption:
  - It seems odd to me to create an Appendix just to duplicate one figure from the main text with an additional row. It would make more sense and be easier for readers to simply add the panels showing the residuals to Fig. 7 and then refer back to that figure in this section. Some discussion of the residuals could be added where Fig. 7 is first presented as well.
  - respective residuals between GLORIA and EMAC --> corresponding residuals (GLORIA minus EMAC)
- p22L12: delete comma after "region"
- p22L13: These findings about the impact of scavenging by high-altitude cirrus on $HNO_3$ in the UT/LMS should be placed in the context of other studies that have examined this issue.

Section 5: Discussion and Conclusions:
- p23L2: What does "ACM" mean? Also: during --> taken during
- p23L13: delete "used"
- p23L15: by generated cloud masks from --> by cloud masks generated from
- p23L17: between the models are reproduced to --> between the two models are attributed to
- p23L18: It is not clear what "limitation of the comparison" means here.
- p23L19: respective --> corresponding; used for --> used as
- p24L6: life time --> lifetime
- p24L7: with comparing --> by comparing
- p24L9: 2019) and suggests --> 2019), which suggests
- p24L13: a change in --> a reduction in
- p24L16: show practically --> has practically
- p24L20: Again, "schematically" is not quite the right word here. Maybe "in a broad sense"?
- p24L21: simulations --> simulation
- p24L23-24: "continuous" is not an appropriate word here – aircraft measurements are not continuous. Also: to continuously test --> to continue to test; delete comma after "required"
- p24L22-25: The authors "speculate" that the biases and sensitivities found in this study might help provide better forecasts and long-term projections. But it is not clear to me that they have provided "actionable" information that will really inform model development / refinement in a concrete way. It might help to add another sentence or two about how they think these results could be used to guide model improvement efforts.

Appendix A:
- p25L8: EMAC-model (panels g-i) between −10 --> EMAC (panels g-i) model at various times between −10
- p25L9: add comma after "geolocations"
- p26L2: and it is --> and is

- p26L3: the measured cloud system by --> the cloud system measured by
- p26L5: is dissolving --> dissolves; "supposably" is not an English word, and I cannot even guess what the authors may have meant so I am unable to offer an alternative ("supposedly" is a word but does not make sense in this context)
- p26L6-7: The cloud system not only appears in the model a few hours earlier than observed but it also covers a much shallower altitude domain.  Is that because of the problem with "false" GLORIA cloud detections below optically dense cloud layers discussed in Section 4.1?  On the other hand, EMAC also shows the cloud to have a much larger vertical extent than ICON-ART.
- p26L6-7: data, however --> data; however,
- p26L10: It is stated that the cloud "breaks apart into two pieces" at T=−6 h, but to me it seems that even at T=−10 h (Fig. A1g) there were already two connected but distinct features.
- p26L10-11: is also dissolving --> dissolves; is also subsiding and decreasing --> subsides and decreases
- p26L14: Figure --> Figures
- p27L1: add a comma after "flight"
- p27L7: delete comma after "cases"
- p27L10: in accordance to --> in accordance with

Recurring minor wording issues:
- p10L19, p17L2, p21L12: it is not clear what is meant by "late" polar vortex in these lines.  If I understand correctly, then "late-stage", "late-winter", or "aged" would be better than "late".
- p10L24, p11L5, p17L3: backward leg --> return leg
- p12L18, p14L1, p14L6, p14L8, p14L16, p15L16, p22L12, p25L8, p26L8, p27L5, p27L7, p27L9: EMAC-simulation --> EMAC simulation; EMAC-cross section --> EMAC cross section; EMAC-model --> EMAC model; EMAC-data --> EMAC data (i.e., delete hyphens)
- p12L18, p14L14, p15L5, p15L6, p15L13, p15L17, p15L23, p25L8, p26L2, p26L14, p27L3: ICON-ART- --> ICON-ART (i.e., delete hyphens after "ART")
- p14L12, p14L14, p14L15, p14L16: GLORIA- --> GLORIA (i.e., delete hyphens)
- p14L27, p15L15, p15L22: at the day --> on the day
- p16L15, p17L1, p17L7: behind --> after
- p17L6, p18L3, p19L11: less details --> fewer details
- p18L4, p18L12, p18L13, p21L21: delete the comma after "al."
- p18L10, p18L21, p19L8, p21L20: hardly --> barely, or, not well

---

## Author Comment (AC1)

**Answer to Comment by Referee 1 20.12.2021**

We thank the Referee 1 very much for his/her time and valuable comments, which helped a lot to improve the manuscript. In the following, we provide our answers to each of the comments and corrections in sequential order. The original Referee comment is repeated in bold, and our answers and changes in the manuscript are provided in italic. Text added to the manuscript is indicated in *blue italic*.

The paper of Haenel et al. presents a comparison of observed 2D distributions of species from the GLORIA instrument with model simulations. The authors use one particular flight from winter 2015/2016 at high northern latitudes (POLSTRACC) to compare the capabilities of ICON-ART and EMAC to simulate H2O, O3 and HNO3 as well as cloud occurrence in the UTLS region. The selected flight comprises very different meteorological situations which allows to evaluate different aspects of the relevant model parametrisations.

The ICON-ART data are based on a R2B6 global simulation with a R2B7 nest in the region of interest, the latter corresponding to 20 km horizontal spacing. EMAC data are available at T106 spectral resolution corresponding to a grid spacing of approx. 40 km at 70N. Data are interpolated at the tangent points of the observations and vertical cross section of relevant species are analysed.

Discrepancies are found for cloud occurrence in ICON-ART. Stratospheric water vapour is simulated too high for EMAC not too surprisingly underestimating the vertical gradients. Contrary, ozone is represented well in EMAC while ICON-ART ozone data suffer from the modified LINOZ-scheme.

The authors put a strong focus on the potential reasons for the misrepresentation of clouds in the high resolution simulation of ICON-ART and conclude on matching / timing problems. For EMAC the applied cloud mask better fits the observations, which the authors partly attribute to the lower resolution (noting fundamental model and diagnostic differences). Further, based on T42-simulations of EMAC they show, that the model resolution plays a key role for the H2O gradients and mixing ratios as well as HNO3 in EMAC. To check the impact of scavenging on HNO3, which is only provided by EMAC, they conclude, that scavenging is essential to simulate HNO3 correctly.

We thank Referee 1 for the precise summary of our study.

The paper is well written, and illustrates some problems of state-of-the-art models to simulate the composition of the challenging UTLS-region governed by strong gradients and often sub-grid processes. However, the central goal of the study is not clear, despite the authors state: " ...with the goal to aid model development and improving our understanding of processes in the upper

**troposphere/lowermost stratosphere...". It leaves the reader with the main key messages: Resolution matters, chemistry matters which are both not too surprising.**

We agree that the goal of the study with regard to aiding model development needs further clarification. Under consideration of the comment by the Referee and comments by Referee 2, we now summarise suggestions for model improvement more clearly in the new Section 4.5 "Suggestions for model improvement". Furthermore, a corresponding summary statement has been added to the Section 5 (Discussion and conclusions).

Following the comment by the Referee given below, we furthermore put a stronger focus on the capabilities of the models of simulating dynamical structures of troposphere-to-stratosphere exchange in the presented case study. As suggested, we now investigate the development of the narrow tropospheric filaments seen in the observations and model data between 17:30 and 18:30 UTC with the help of the model data and discuss the results in Section 4.3. Using ICON-ART, we show that a larger filament in the west was transported horizontally into the Arctic LMS in connection with poleward breaking of a cyclonically sheared Rossby wave, while two weaker filaments in the east are associated with an older tropopause fold there. From our point of view, it is remarkable that the model representation of the underlying processes and the resulting modelled structures result in such a high degree of agreement with the observations.

**Since the fundamental properties of the model systems are very different, but the resolution is one key aspect of the comparison results the authors should provide in addition a comparison of similar grid spacing (e.g. between T106, R2B6 or coarse graining).**

We agree that a comparison of the different models in a similar resolution would allow for a more direct comparison of EMAC and ICON-ART. Following the suggestion by the Referee, we revised Figure 5 and 7 and added a new Figure to analyse differences between the ICON-ART global and nested domains. However, the different properties of the grids should be kept in mind in the comparison (cf. Fig. 2 and 3), and the EMAC grid does not converge towards the poles in meridional direction.

Specifically, we included an additional panel with the ICON-ART cloud mask at the R2B6 grid in Figure 5, which shows only small differences between the ICON-ART global and nested domains.

Furthermore, we replaced the results of the ICON-ART nested R2B7 domain in Figure 7 by the global ICON-ART R2B6 domain to allow a more direct comparison with EMAC (see below). Following the suggestion by Referee 2, we furthermore included residual plots between the corresponding model and GLORIA cross sections.

In a new Figure (Fig. 8, see below), we investigate residuals between the ICON-ART global and nested domain and provide a bridge from the ICON-ART global domain to the nested domain used in the following. We added the following discussion:

"To investigate potential differences between the global R2B6 and the nested R2B7 ICON-ART domain, differences between these grids are depicted in Fig. 8. Mesoscale patterns in the residuals of  $q_v$  (Fig. 8a) and  $O_3$  (Fig. 8b) in the tropopause region and, in the case of  $q_v$ , in the regions where clouds were present (compare Fig. 5), are attributed to finer/coarser representation by the different

model grids and the subsequent interpolation to the GLORIA geolocations. Overall, no significant systematic biases are identified."

---

## Author Comment (AC2)

**Answer to Comment by Michelle Santee (Referee 2)**

20.12.2021

We thank the Michelle Santee very much for her time, valuable comments, and detailed corrections, which helped a lot to improve the manuscript. In the following, we provide our answers to each of the comments and corrections in sequential order. The original Referee comment is repeated in bold, and our answers and changes in the manuscript are provided in italic. Text added to the manuscript is indicated in *blue italic*.

**Review of "Challenge of modelling GLORIA observations of UT/LMS trace gas and cloud distributions at high latitudes: a case study with state-of the-art models" by Haenel et al.**

**This manuscript uses GLORIA measurements taken on a PGS flight that sampled a diverse set of conditions in the UT/LMS to test the ability of two models, ICON-ART and EMAC, to simulate cloud structures and trace gas ($H_2O$, $O_3$, $HNO_3$) distributions. Both models are shown to reproduce the observations quite well; discrepancies between modelled and measured cloud and composition fields are quantified and their causes investigated. The paper is well organized and well written, and the figures are generally well prepared and support the discussion. I have only a few substantive issues that I would like to see addressed before the paper is accepted for publication; most of my comments are minor wording suggestions that will take very little time to act on.**
**Below both major substantive issues and minor points of clarification, wording suggestions, and grammar / typo corrections are listed together for each Section in sequential order.**

**Respectfully,**
**Michelle Santee**

*We thank Michelle Santee for the accurate summary and appreciate the positive rating.*

**Abstract:**
- **p1L18-19: The wording "measurements taken in a challenging case study by the GLORIA" could be interpreted to imply that that flight was deliberately designed for this purpose, which I do not believe was the case. I think it would be better to simply say "measurements taken in a flight of the GLORIA" here and then add "challenging" in front of "multifaceted" in L23.**
  *Agreed and done*
- **p1L21: 2016, which --> 2016 that**
  *Done*
- **p2L3: moist-bias --> moist bias**
  *Done*
- **p2L7: changing of the --> changing the**

*Done*

- **p2L8: play only a role in case of HNO₃ --> play a role only in the case of HNO₃**

  *Done*

- **p2L8-9: I agree that the representativeness of these results is an important question that should be explored. However, unless I missed it, this issue is not raised anywhere in the paper other than this sentence in the abstract. It should be acknowledged elsewhere as well, at least in the Discussion and Conclusions section if not somewhere in the main text of the paper.**

  *We agree that the representativeness of the results of our study should be addressed. We revisited the abstract and came to the conclusion that the corresponding statement there does not provide much helpful information. Therefore, we rephrased this part of the abstract and address the representativeness of our study now in more detail in the discussion and conclusions (Section 5):*

  *"The GLORIA data were measured during a single flight on 26 February 2016 with a duration of 9 hours 40 minutes and a total distance of ~8000 km. The flight covered a multifaceted scenario of the UT/LMS at high latitudes performed prior to the final major warming (Manney and Lawrence, 2016, and Matthias et al., 2016). Therefore, the presented comparisons of the GLORIA and model data can be considered representative for the polar UT/LMS at high latitudes in late winter prior to the vortex breakdown."*

- **p2L10: projection --> projections**

  *Done*

- **p2L10: Although this study has certainly provided very useful information to characterize model biases, I am less convinced that it has really laid out sufficiently specific guidance to "define paths for further model improvements". See related final comment on the Conclusions below.**

  *We agree that this wording might has overrated a bit the outcome of our study. We changed the wording to "…and provide suggestions for further model improvements." We furthermore agree that suggestions for model improvement should be summarized and discussed more clearly. We now summarize the observed model biases and provide suggestions for model improvement in the new Section 4.5 "Suggestions for model improvement" (see reply to Referee 1). Furthermore, a corresponding summary statement has been added to discussion and conclusions (Section 5).*

**Section 1: Introduction:**

- **In a number of places in the presentation of background material (e.g., p2L13, p2L16, p2L21, p2L27, p2L28, p3L7), a few citations are given for very well-established concepts, but many other equally suitable papers could have been cited instead of or in addition to the ones listed. Obviously not all relevant papers can be referenced for these points, but "e.g." should be added in these lines to avoid giving readers the impression that the selected references are the only appropriate ones.**

  *Agreed and done*

  **p2L17-18: spread in these trends among models while perturbating ozone and other greenhouse gas abundances --> spread among modelled trends when ozone and other greenhouse gas abundances are perturbed**

*Done*

- **p2L18: can be --> include**
  *Done*
- **p2L21: knowledge on --> knowledge of**
  *Done*
- **p2L23: compartment --> layer**
  *Done*
- **p2L24: On the winter --> In the winter**
  *Done*
- **p3L6: sedimentation … redistribute --> sedimentation … redistributes; "eventually" is not really needed here, but if the authors want to keep it, it should come before "changes"**
  *Agreed and done*
- **p3L12: I do know what is meant by "(in parts)" in this sentence; if I have understood the intent, this would be better as: "in part explicitly and in part by using" or "both explicitly and by using"**
  *We have changed the sentence to* "in part explicitly and in part by using".
- **p3L13: such models are the models ICON --> such models include ICON**
  *Done*
- **p3L34: I assume that the systematic biases referred to here are in the model fields (that is, the intention is not to use the model results to validate the GLORIA data), but that should be made explicitly clear, e.g.: in the trace gas distributions --> in the modelled trace gas distributions**
  *Agreed and done*

**Section 2: Data and diagnostics:**
- **p4L10: the used model setups --> the model setups used**
  *Done*
- **p4L11: overview on --> overview of**
  *Done*
- **p4L15: aircrafts --> aircraft**
  *Done*
- **p4L24: combined to --> combined into**
  *Done*
- **p5L7: is operational --> has been operational**
  *Done*
- **p6L16: delete the comma after "winter"**
- *Done*
- **p6L19: air masses suitable for --> air masses whose conditions are conducive to**
  *Done*
- **p7L1: life time --> lifetime**
  *Done*
- **p7L9: does "between ~12 to 21 hours" mean "between ~12 and 21 UTC"?**

*We have changed the sentence to "forecasts with lead times of ~12 to 21 hours", as we meant the lead time (i.e. the running time) of the forecast. Here, this time is identical with the time points during the flight, since the model data is interpolated in space and time to the GLORIA geolocations. For clarification, we added: "(depending on point in time during flight)"*

- **p7L16: atmosphere --> atmospheric**
  *Done*

- **p7L20: in the --> with; T106L90MA-resolution --> T106L90MA resolution**
  *Done*

- **Fig. 3 caption: The corresponding T106 (T42) grid corresponds --> The T106 (T42) grid corresponds; reduces --> is reduced**
  *Done*

- **p7L23-25: This is a very awkwardly worded and unclear sentence. If I have understood it correctly, it would be clearer to say: "To simulate realistic synoptic conditions, surface pressure and various prognostic variables (temperature, vorticity, and divergence) are "nudged" towards the ECMWF ERA-Interim reanalysis (Dee et al., 2011) above the boundary layer and below 1 hPa using a Newtonian relaxation technique." This formulation of the sentence also introduces the term "nudged", which is used later in the manuscript but is not currently defined.**
  *Agreed, thanks. We adapted the suggested wording.*

- **p7L26: a comprehensive chemistry --> a comprehensive chemistry scheme**
  *Done*

- **p7L27-30: Because of the complexity of the punctuation in these lines, some of the commas need to be replaced with semicolons (marked in red here): "… (Sander et al., 2011); the photolysis submodel JVAL (Sander et al., 2014); the submodel MSBM, mainly responsible for the simulation of PSCs (Kirner et al., 2011); the submodel CLOUD, based on the ECHAM5 cloud scheme, simulating large scale clouds (Roeckner et al., 2006); the submodel CONVECT, calculating convection and convective clouds (Tost et al., 2006b); and …".**
  *Done*

- **p9L9: combined to --> combined into**
  *Done*
  **p9L17: delete "that deviate"**
  *Done*

- **p9L18: e.g. --> i.e.**

- *Done*

**Section 3: Flight overview and meteorological analysis:**
- **p10L2: concerning the decades before --> relative to preceding decades**
  *Done*

- **p10L6: ended by --> ended with**
  Done

- **p10L9 and L11: take off --> takeoff**

*Done*

- **p1011-12: headed westwards (GLORIA pointing to northward directions) --> headed westward (GLORIA pointing northward)**
  *Done*

- **p10L12-13: turned to a southward direction (GLORIA pointing to westward directions) --> turned southward (GLORIA pointing westward)**
  *Done*

- **p10L13-14: back to eastward directions and … southwards) --> back eastward and … southward)**
  *Done*

 • **Fig. 4 and its caption:**
   ○ **It is very difficult to make out the overlaid white contours without greatly magnifying the plot – thicker lines and larger font for the labels would make them easier to see.**
   *Done*

   ○ **It would be convenient to have waypoints A and B marked on these maps as well as on Fig. 1.**
   *Done*

   ○ **is colour-coded in contour --> is shown by colour contours; delete "together"; occlusions --> occlusions (black overlays)**
   *Done, (we furthermore added "(black and dark grey overlays)", since further fronts and another occlusion above the Atlantic are shown in panel (d) in dark grey)*

- **p10L21: west- --> westward-; way-point --> waypoint**
  *Done*

- **p10L24: partly dissipates on --> has partly dissipated by**
  *Done*

- **p10L24-25: pointed subsequently towards … and into --> pointed first towards … and then into**
  *Done*

- **p10L29: the wording "going along with" is not completely clear.  Does this mean "consistent with" or "accompanied by"?**
  *We meant "accompanied by". We corrected the wording accordingly*

- **p10L30-31: move "to date" to after "on record"**
  *Done*

- **p11L6: Why is the word "subsequently" used here?  The air masses observed during the flight contained these features – they were not observed subsequent to the flight.**
  *Agreed. We deleted the word "subsequently".*

**Section 4: Observed and modelled cloud and trace gas distributions:**

- **p12L4-5: It is stated here that CI values "approaching four and higher" are indicative of cloudfree conditions.  Since the color scale in Fig. 5a saturates at CI=3.0, does that imply that on this flight GLORIA never encountered air masses that can be considered cloud-free?**

*Thanks for pointing out this inconsistency. The threshold value of 4 applies to spaceborne limb-sounding observations, while different threshold values were found to be suitable for airborne limb sounding. We added the following explanation:*

*"In the case of airborne limb observations, CI values of 2 to 4 have been found to be suitable to separate between cloud-affected and cloud-free conditions in previous studies (Johansson et al., 2018 and references therein). In the case presented here, a cloud index of ~2.5 represents the threshold between cloud-affected and cloud-free conditions."*

- **p12L9: It seems a bit odd to characterize air masses affected by clouds as having an "enhanced" CI since it is actually low values of CI that indicate the presence of clouds.**
  *Agreed, we changed the wording to "reduced".*
- **p12L15: used cloud masks --> cloud masks used**
  *Done*
- **p12L18: threshold of the cloud mask for the ICON-ART- and EMAC-model at --> threshold for the ICON-ART and EMAC model cloud masks at**
  *Done*
- **p12L22: add a comma after "concentrations"**
  *Done*
- **Fig. 5b,c: I am a bit confused about why the tick marks on the cloud mask color bars are needed**
  *Agreed. We removed the tick marks*
- **p12L33: Discrepancies between measured and modelled clouds at lower altitudes for the system around 20 UTC are attributed to GLORIA data being affected by optically dense cloud layers above, but couldn't this explanation be applied to the mismatch for other clouds as well, such as the one between 12 and 13 UTC?**
  *Yes, agreed. We now mention in the text that the same effect might explain the differences between the observed and modelled cloud systems between 12 and 13 UTC at lower altitudes.*
- **p13L1: delete the comma after "GLORIA"**
  *Done*
- **p13L3: delete the comma after "fact"**
  *Done*
- **p13L8: I'm not sure what "appear more sharply in the ICON-ART simulation" means, as the cloud systems in question barely register at all in the model cloud mask.**
  *Agreed. We changed the sentence to "…are barely reproduced in the ICON-ART simulation"*
- **p13L10: respective --> corresponding; EMAC-standard simulation (STD) --> EMAC standard simulation (EMAC-STD); T106L90MA-resolution --> T106L90MA resolution**
  *Done*
- **p14L1: Recalling that --> As mentioned earlier; 2.3), however the --> 2.3); however, the**
  *Done*
- **p14L2: the EMAC standard simulation (STD) --> the EMAC-STD**
  *Done*
- **p14L11: better comparable --> more comparable**
  *Done*

- **p14L19: delete comma after "model"**
  *Done*
- **p14L21: to which degree --> the degree to which**
  *Done*
- **p14L25: does "~12 h to ~20 h" mean "~12 UTC to ~20 UTC"?**
  *In this case we mean the lead time (i.e. running time) of the forecast, which is however identical here with times of the GLORIA geolocations. For clarification, we added "forecast lead time" prior to "between".*
  **Also: accumulated --> cumulative**
  *Done*
- **p15L4-5: It is stated that "all of the observed cloud systems coincide qualitatively with a corresponding precipitation pattern at the respective geolocations in the ICON-ART-data", but the Δ H2O diagnostic does not pick up the cloud system observed by GLORIA prior to 12 UTC.**
  *We agree that there are only very weak indications in the ΔH2O diagnostic prior to 12 UTC (i.e. at ~11:45 and 11:55). We added: "However, as in the case of the ICON-ART cloud mask prior to 12 UTC, only weak indications of cloud systems are found here."*

- **p15L8: Although I see weak negative residuals just below the tropopause, even with the figure greatly magnified it is difficult to discern non-negligible residuals above the tropopause.**
  *Agreed. We changed the range of the colour scale to -10 … 10 ppmv to make it easier to identify these patterns:*

[Figure]

***Figure 6.*** *Modelled short-term changes in specific humidity due to cloud processes. Residuals between nested ICON-ART domain of specific humidity and corresponding $H_2O$ tracer without cloud microphysics. Black dashed lines: ICON-ART 2 PVU and 4 PVU isolines (lower and higher lines, respectively) as indicators for the dynamical tropopause. Grey lines: HALO flight altitude.*

- **p15L10-11: vicinity is found at 14 UTC and reaches --> vicinity at 14 UTC reaches**
  *Done*
- **p15L11: support that --> support the idea that**
  *Done*
- **p15L18: again, hints --> again hints**
  *Done*
- **p15L20: beside --> in addition to**
  *Done*

- **p16L3-4: The use of the term "precipitation" is ambiguous here – I believe that the authors mean "cirrus cloud ice particle sedimentation", but that should be clarified. I also think that it would be appropriate to add discussion putting these results about "precipitation" affecting the humidity of the LMS into the context of previous studies that have examined the impact of convection and cirrus cloud processes on moistening / dehydrating the LMS (especially in light of my previous comment that I had trouble identifying these weak signatures in Fig. 6).**
  *Agreed, we meant "cirrus clouds ice particle sedimentation" and corrected the text accordingly. We rephrased p14/L19ff as following:*
  *"Another proxy for the characterisation of detectable cloud systems in the model is looking at the cirrus cloud ice particle sedimentation events, which include the processes of nucleation, sedimentation and subsequent evaporation of cirrus cloud ice particles. As a consequence, local irreversible dehydration is found when ice particle growth removed water from the gas phase, and hydration is found at lower altitudes where the particles sublimate."*
  *To provide context to previous studies, we added the following statement after p16L4:*
  *"Cirrus clouds under cold conditions in the LMS have been found by many observations (e.g. Lelieveld et al., 1999; Kärcher and Solomon, 1999; Spang et al., 2015) and are likely to affect LMS humidity by ice particle sedimentation (e.g. Kärcher, 2005). Furthermore, as discussed in the literature, convective hydration is known to affect the LMS and can drive air masses to saturation (Schoeberl et al., 2018; Zou et al., 2021)."*
- **p16L4: affects also significantly --> also significantly affects**
  *Done*
- **p16L5: the major cloud systems --> the major cloud systems observed by GLORIA**
  *Done*
- **p16L7: the ICON-ART lacks the simulation of the --> ICON-ART fails to simulate the**
  *Done*
- **p16L11: add a comma after "q$_v$"**
  *Done*
- **p16L12: It should be reiterated when Fig. 7 is introduced that the presence of optically thick clouds precludes trace gas retrievals, as comparison of the patterns in Figs. 5 and 7 shows.**
  *Agreed. We added: "When compared with the cloud index plot (Fig. 5a), gaps in the retrieved trace gas distributions are explained by the fact that the presence of dense clouds precludes trace gas retrievals in the affected regions. Cloud filtering is applied here prior to the trace gas retrieval."*
- **p16L13-14: The tropopause is located near 10 km in all panels of Fig. 7, not just 7a. In addition, use either "around" or "~", not both (see also L16).**
  *Done*
- **p16L17: south-western --> southwestern; part --> flight segment**
  *Done*
- **p16L18-19: reach by ~2km up into --> reach as far as ~2km into**
  *Done*
- **p16L21: complimentary --> complementary (but "converse" is probably a better word here)**

*Agreed, we have changed the word to* *"converse"*

- **p16L25: reach up --> reach nearly up; add a comma after "altitude"**
*A closer inspection of the Fig 7b shows that the filaments in ozone reach even higher up to the flight altitude. We therefore inserted* *"even"* *instead of "nearly" and modified the end of the sentence to* *", therefore deeper into the LMS than the filaments seen in the water vapour distribution."* *We added the comma, as suggested.*

- **p16L27-29: Although I don't doubt that some nitrification at lower altitudes occurred during this winter, the morphology of the $HNO_3$ distribution (Fig. 7c) does not seem very different from that of $O_3$ (Fig. 7b) to me, and abundances of both would be expected to be higher in the LMS than in the UT. Thus I am not certain what local maxima in $HNO_3$ are being referred to here. The specific signatures of nitrification in this figure should be clarified.**
*Thanks for pointing out this shortcoming. We agree that the local maximum can hardly be seen with the applied range of the colour bar and the discussion is unclear. We extended the colour bar to 8 ppbv to resolve the maximum more clearly. Furthermore, there was a mistake in the time interval. Our intention is to discuss the local $HNO_3$ maximum below flight altitude between 15 UTC and ~17 UTC seen in the updated plot, which is not found in the $O_3$ distribution.*
*Furthermore, Ziereis et al. (2021) discuss in their recent publication that during this flight both nitrified air masses, but also denitrified air masses that had descended from above were probed. We corrected the sentence with regard to the time interval of the local maximum. Furthermore, with reference to Ziereis et al. (2021), we now discuss that both, nitrified air masses (prior to ~15 UTC and after ~17 UTC) and denitrified air masses (between ~15 and 17 UTC) were probed at flight altitude. We discuss furthermore that the local maximum seen in the GLORIA data below flight altitude between 15 and 17 UTC is interpreted as subsided nitrified air masses which are located below denitrified air masses at flight altitude in this section of the flight.*
*This interpretation is furthermore consistent with the results of the EMAC sensitivity run, which shows nitrification below flight altitude between 15 and 17 UTC (Fig. 10c, now Fig. 12c).*

- **p17L8: is the comparison ("higher") with respect to ICON-ART or GLORIA? Assuming the latter: reach here higher up by 1-2 km --> reach altitudes higher than those observed by 1-2 km**
*Yes, we meant with respect to GLORIA. We modified the sentence accordingly*
**p17L12: "schematically" is not quite the right word here – maybe "broadly" or "generally"?**
*Agreed, we have changed the wording to* *"broadly"*

- **p18L3: fine-structures --> fine structures**
*Done*

- **p18L19: complimentary --> complementary**
*Done*

- **p18L31: Thereby --> However**
*Done*

- **p18L32: which is by a factor of ~5 lower than that the ICON-ART R2B7 nest --> which is lower than that of the ICON-ART R2B7 nest by about a factor of 5**
*Done*

- **p19L2: delete "respective"**

*Done*

- **p19L3-4: It would be appropriate to acknowledge some of the previous studies that have also found substantial troposphere-to-stratosphere exchange associated with tropopause fold events; folded airmass structures reach --> airmasses in tropopause fold structures reach**

  *We added references regarding tropopause folding and air mass exchange. We adapted the wording suggested by the Referee with a slight modification (i.e., "variations in the dynamical tropopause"), since a less developed tropopause fold is found here when compared with the study by Shapiro (1980). We rephrased as follows:*

  *"The combination of ozone and water vapour data clearly shows that air masses characterised by tropospheric moisture levels reach deeply into the LMS and are connected to variations in the dynamical tropopause. Tropopause folds and steps in the tropopause are regions where isentropic levels cross the tropopause and jet streams. They are known bidirectional exchange regions between the tropopause and stratosphere (e.g. Shapiro, 1980; Keyser and Shapiro, 1986) and to contribute to transport and mixing of tropospheric air into the LMS such as diagnosed e.g. by Werner et al. (2010), Krause et al. (2018), and Jing et al. (2018) (note however that a net exchange from the LMS to the troposphere dominates)."*

  *Following the suggestion by Referee 1, we added a detailed analysis of the evolution of the filaments and tropopause folds at the days before the flight (see Reply to Referee 1).*

- **p19L6: shows highly --> shows a highly**

  *Done*

- **p19L7: I think that "broadly captures" or something like that would be better wording than "resolves in principle"**

  *Done*

- **p19L10: In case of --> In the case of**

  *Done*

- **p19L14: by both --> from both**

  *Done*

- **p20L5-8: I think that the flow would be improved by moving the introduction of Fig. 10 in these lines to after the end of the discussion of Fig. 9 on the following page.  Also: on it) --> on them)**

  *Agreed and done*

- **p20L11: bias, which is known for the --> bias that is known to affect the**

  *Done*

- **Fig. 9 caption: EMAC and ICON-ART output --> ICON-ART and EMAC output**

  *Done*

- **p21L5: found and increases --> found that increases**

  *Done*

- **p21L8: fine-structures --> fine structures**

  *Done*

- **Fig. 10: It might be helpful to add an overlay outlining the zero contour, especially in Fig. 10b, since it is hard to tell where the EMAC ozone residuals change sign.**

  *Done*

- **p21L12: that ozone --> that the ozone**

  *Done*
- **p21L14: B, ozone is significantly --> B is ozone significantly**

  *Done*
- **p21L17: scheme by --> scheme used by**

  *Done*
- **p21L18-19: above the troposphere and strongly --> above the tropopause that strongly; amounts --> amounts to**

  *Done*
- **p21L21: while comparing --> in a comparison of**

  *Done*
- **p21L23: due --> due to; (de-) nitrification --> denitrification/nitrification**

  *Done*
- **p21L23-24: It is confusing to focus only on the evaporation of PSC particles here, as that leads to HNO$_3$ enhancement (renitrification). If I understand correctly, the modelled HNO$_3$ depletion associated with the subsided air mass encountered in the middle of the flight is being attributed to sequestration in existing PSC particles or permanent denitrification through their subsequent sedimentation. That should be clarified.**

  *We agree that this sentence is confusing. As discussed by Ziereis et al. (2021), denitrified air masses are seen in the middle of the flight (as discussed by the authors, PSC particles were not detected any more in situ at flight altitude during this phase of the winter). At higher altitudes, sequestration in existing PSC particles might still have played a role here, if temperatures were cold enough. We corrected the sentence accordingly.*
- **Fig. 11 caption: T106 vs T42 resolution --> T106 minus T42 resolution**

  *Done*
- **p22L1-5: I am not convinced of the value of including the T106 vs T42 sensitivity test shown in Fig. 11a-c, as the benefit of using the higher resolution in EMAC has already been demonstrated in the Khosrawi et al. papers mentioned here. Why was it necessary to repeat this comparison?**

  *We agree that the conclusion is the same as provided by Khosrawi et al. However, we think that our results are still useful, since the study by Khosrawi et al. focused mainly on the stratosphere, while our study has a more detailed focus at the upper UT/LMS region. Furthermore, our study provides another dataset to support this interpretation. We changed the wording to: "A similar behaviour of EMAC was found in the stratosphere by Khosrawi et al. (2017), who stated …"*
- **p22L2: "enhances" can have a positive connotation, hence: enhances --> exacerbates**

  *Done*
- **p22L4: stating --> who stated; MLS --> Microwave Limb Sounder**

  *Done*
- **p22L6: The findings about scavenging processes only being important for HNO$_3$ are presented here and later in Section 5 in a manner that suggests that they were unexpected. Did the**

**authors have any expectation that scavenging processes would affect the O₃ or H₂O distributions? More background and context motivating this sensitivity test is needed.**

*We agree that these aspects should be addressed and added the following statement:*

*"Scavenging processes by cirrus cloud ice particles are capable of removing trace gases from the gas phase. Sedimentation of the ice particles is capable of removing the trapped gases from affected altitudes. While previous studies focused mainly on scavenging on liquid cloud droplets (Tost et al., 2010; Wang et al., 2010; Pierce et al., 2015; Kaiser et al., 2019), Tost et al. (2010), however, found HNO₃ values in the Northern hemisphere upper troposphere to be low due to uptake on ice particles and subsequent sedimentation. Thereby, relative changes were found to be large due to low absolute values there. In addition, the vertical redistribution of HNO₃ could induce secondary effects on other trace gases via chemical processes. In particular, altering HNO₃ could lead to changes in the budget of reactive nitrogen oxides (NOₓ), which, in turn, could impact ozone (e.g. Kelly et al., 1991; Krämer et al., 2008; Schiller et al., 2008). Here, our goal is to test whether the effect of scavenging over ice on the trace gas composition is significant in the LMS in the EMAC simulation."*

- **p22L7: ppbv in --> ppbv than in**
  *Done*

- **p22L8: Reminding --> Recalling**
  *Done*

- **p22L10: delete ", however,"; most parts of a region --> most of the region**
  *Done*

- **p22L11: delete "respective"; delete comma after "means"**
  *Done*

- **Fig. B1 and caption:**
  - **It seems odd to me to create an Appendix just to duplicate one figure from the main text with an additional row. It would make more sense and be easier for readers to simply add the panels showing the residuals to Fig. 7 and then refer back to that figure in this section. Some discussion of the residuals could be added where Fig. 7 is first presented as well.**
    *Agreed. We added the residuals to Fig. 7 and discuss them already in this section.*
  - **respective residuals between GLORIA and EMAC --> corresponding residuals (GLORIA minus EMAC)**
    *Done*

- **p22L12: delete comma after "region"**
  *Done*

- **p22L13: These findings about the impact of scavenging by high-altitude cirrus on HNO₃ in the UT/LMS should be placed in the context of other studies that have examined this issue.**
  *See comment to p22/L6. We furthermore added a statement that our results are consistent with the results by Tost et al. (2010), who found a similar effect in the upper troposphere.*

**Section 5: Discussion and Conclusions:**

- **p23L2: What does "ACM" mean?  Also: during --> taken during**
  *We have now spelled out "ACM" to "atmospheric chemistry model"*
- **p23L13: delete "used"**
  *Done*
- **p23L15: by generated cloud masks from --> by cloud masks generated from**
  *Done*
- **p23L17: between the models are reproduced to --> between the two models are attributed to**
  *Done*
- **p23L18: It is not clear what "limitation of the comparison" means here.**
  *Our goal was to express that the comparison of the measured quantity cloud index with cloud masks generated from the models is limited. We modified the sentence accordingly.*
- **p23L19: respective --> corresponding; used for --> used as**
  *Done*
- **p24L6: life time --> lifetime**
  *Done*
- **p24L7: with comparing --> by comparing**
  *Done*
- **p24L9: 2019) and suggests --> 2019), which suggests**
  *Done*
- **p24L13: a change in --> a reduction in**
  *Done*
- **p24L16: show practically --> has practically**
  *Done*
- **p24L20: Again, "schematically" is not quite the right word here.  Maybe "in a broad sense"?**
  *Agreed, we have changed the wording to "broadly"*
- **p24L21: simulations --> simulation**
  *Done*
- **p24L23-24: "continuous" is not an appropriate word here – aircraft measurements are not continuous.**
  *Agreed, we have meant "continuing", done*
  **Also: to continuously test --> to continue to test; delete comma after "required"**
  *Done*
- **p24L22-25: The authors "speculate" that the biases and sensitivities found in this study might help provide better forecasts and long-term projections.  But it is not clear to me that they have provided "actionable" information that will really inform model development / refinement in a concrete way.  It might help to add another sentence or two about how they think these results could be used to guide model improvement efforts.**
  *Agreed, see comment to p2L10.*

**Appendix A:**

- **p25L8: EMAC-model (panels g-i) between −10 --> EMAC (panels g-i) model at various times between −10**

  *Done*

- **p25L9: add comma after "geolocations"**

  *Done*

- **p26L2: and it is --> and is**

  *Done*

- **p26L3: the measured cloud system by --> the cloud system measured by**

  *Done*

- **p26L5: is dissolving --> dissolves; "supposably" is not an English word, and I cannot even guess what the authors may have meant so I am unable to offer an alternative ("supposedly" is a word but does not make sense in this context)**

  *Agreed, we have changed the wording to "presumably"*

- **p26L6-7: The cloud system not only appears in the model a few hours earlier than observed but it also covers a much shallower altitude domain.  Is that because of the problem with "false" GLORIA cloud detections below optically dense cloud layers discussed in Section 4.1? On the other hand, EMAC also shows the cloud to have a much larger vertical extent than ICON-ART.**

  *In principle, the explanation with regard to optically thick cloud tops might partly explain the discrepancy here, too. However, since some structures are seen in the GLORIA cloud index at lower altitudes here, this cloud, at least in parts, are not completely optically thick. Furthermore, it should be remembered that the comparison of the model cross sections several hours before the measurements is limited, since the atmospheric scenario changes. We added these aspects in the discussion.*

- **p26L6-7: data, however --> data; however,**

  *Done*

- **p26L10: It is stated that the cloud "breaks apart into two pieces" at T=−6 h, but to me it seems that even at T=−10 h (Fig. A1g) there were already two connected but distinct features.**

  *Agreed. We modified the discussion accordingly.*

- **p26L10-11: is also dissolving --> dissolves; is also subsiding and decreasing --> subsides and decreases**

  *Done*

- **p26L14: Figure --> Figures**

  *Done*

- **p27L1: add a comma after "flight"**

  *Done*

- **p27L7: delete comma after "cases"**

  *Done*

- **p27L10: in accordance to --> in accordance with**

  *Done*

**Recurring minor wording issues:**

- **p10L19, p17L2, p21L12: it is not clear what is meant by "late" polar vortex in these lines. If I understand correctly, then "late-stage", "late-winter", or "aged" would be better than "late".**
  *Agreed, changed the wording to "late-stage"*
- **p10L24, p11L5, p17L3: backward leg --> return leg**
  *Done*
- **p12L18, p14L1, p14L6, p14L8, p14L16, p15L16, p22L12, p25L8, p26L8, p27L5, p27L7, p27L9: EMAC-simulation --> EMAC simulation; EMAC-cross section --> EMAC cross section; EMACmodel --> EMAC model; EMAC-data --> EMAC data (i.e., delete hyphens)**
  *Done*
- **p12L18, p14L14, p15L5, p15L6, p15L13, p15L17, p15L23, p25L8, p26L2, p26L14, p27L3: ICON-ART- --> ICON-ART (i.e., delete hyphens after "ART")**
  *Done*
- **p14L12, p14L14, p14L15, p14L16: GLORIA- --> GLORIA (i.e., delete hyphens)**
  *Done*
- **p14L27, p15L15, p15L22: at the day --> on the day**
  *Done*
- **p16L15, p17L1, p17L7: behind --> after**
  *Done*
- **p17L6, p18L3, p19L11: less details --> fewer details**
  *Done*
- **p18L4, p18L12, p18L13, p21L21: delete the comma after "al."**
  *Done*
- **p18L10, p18L21, p19L8, p21L20: hardly --> barely, or, not well**
  *Done*

*References*

*Jing, P. and Banerjee, S.: Rossby wave breaking and isentropic stratosphere-troposphere exchange in 1981–2015 in the Northern Hemisphere, J. Geophys. Res.-Atmos., 123, 9011–9025, https://doi.org/10.1029/2018JD028997, 2018.*

*Kärcher, B.: Supersaturation, dehydration, and denitrification in Arctic cirrus, Atmos. Chem. Phys., 5, 1757–1772, https://doi.org/10.5194/acp-5-1757-2005, 2005.*

*Kärcher, B. and Solomon, S.: On the composition and optical extinction of particles in the tropopause region, J. Geophys. Res.,104, 27 441-27 459, 1999.*

*Kaiser, J. C., Hendricks, J., Righi, M., Jöckel, P., Tost, H., Kandler, K., Weinzierl, B., Sauer, D., Heimerl, K., Schwarz, J. P., Perring, A. E., and Popp, T.: Global aerosol modeling with MADE3 (v3.0) in EMAC (based on v2.53): model description and evaluation, Geosci. Model Dev., 12, 541–579, https://doi.org/10.5194/gmd-12-541-2019, 2019.*

Keyser, D. and Shapiro, M.: A review of the structure and dynamics of upper-level frontal zones, Mon. Weather Rev., 114, 452–499, https://doi.org/10.1175/1520-0493(1986)114<0452:AROTSA>2.0.CO;2, 1986.

Lelieveld, J., Bregman, A., Scheeren, H. A., Ström, J., Carslaw, K. S., Fischer, H., Siegmund, P. C., and Arnold, F.: Chlorine activation and ozone destruction in the northern lowermost stratosphere, J. Geophys. Res., 104, 8201–8213, 1999.

Pierce, J. R., Croft, B., Kodros, J. K., D'Andrea, S. D., and Martin, R. V.: The importance of interstitial particle scavenging by cloud droplets in shaping the remote aerosol size distribution and global aerosol-climate effects, Atmos. Chem. Phys., 15, 6147–6158, https://doi.org/10.5194/acp-15-6147-2015, 2015.

Schoeberl, M. R., Jensen, E. J., Pfister, L., Ueyama, R., Avery, M., and Dessler, A. E.: Convective hydration of the upper troposphere and lower stratosphere, J. Geophys. Res.-Atmos., 123, 4583–4593, 2018.

Shapiro, M. A.: Turbulent Mixing within Tropopause Folds as a Mechanism for the Exchange of Chemical-Constituents between the Stratosphere and Troposphere, J. Atmos. Sci., 37, 994–1004, 1980.

Tost, H., Lawrence, M. G., Brühl, C., Jöckel, P., The GABRIEL Team, and The SCOUT-O3-DARWIN/ACTIVE Team: Uncertainties in atmospheric chemistry modelling due to convection parameterisations and subsequent scavenging, Atmos. Chem. Phys., 10, 1931–1951, https://doi.org/10.5194/acp-10-1931-2010, 2010.

Wang, X., Zhang, L., and Moran, M. D.: Uncertainty assessment of current size-resolved parameterizations for below-cloud particle scavenging by rain, Atmos. Chem. Phys., 10, 5685–5705, https://doi.org/10.5194/acp-10-5685-2010, 2010.

Ziereis, H., Hoor, P., Grooß, J.-U., Zahn, A., Stratmann, G., Stock, P., Lichtenstern, M., Krause, J., Afchine, A., Rolf, C., Woiwode, W., Braun, M., Ungermann, J., Marsing, A., Voigt, C., Engel, A., Sinnhuber, B.-M., and Oelhaf, H.: Redistribution of total reactive nitrogen in the lowermost Arctic stratosphere during the cold winter 2015/2016, Atmos. Chem. Phys. Discuss. [preprint], https://doi.org/10.5194/acp-2021-707, in review, 2021.

Zou, L., Hoffmann, L., Griessbach, S., Spang, R., and Wang, L.: Empirical evidence for deep convection being a major source of stratospheric ice clouds over North America, Atmos. Chem. Phys., 21, 10457–10475, https://doi.org/10.5194/acp-21-10457-2021, 2021.

---

## Author Response (AR2)

**Answer to Review by Editor**

**17.01.2022**

We thank the Editor Mathias Palm very much for his time, thorough revision, and helpful comments. In the following, we provide our answers to each of the comments and corrections. The original Editor comment is repeated in bold, and our answers are provided in *italic*. Changes in the manuscript are indicated in *blue italic*.

**Comments to the author:**

Dear authors, many thanks for your work and the detailed reply to the reviews. However there is one point which I do not find convincing:

In your paragraph about the AVK use (page 10 first paragraph), you write, that you do not use AVK's because you do not expect that the situation improves significantly. I find this is an odd statement in the context of your work.

You cite Ungermann (2011) as a justification, that the AVK's would not have an effect. But when I understand Ungermann (2011) correctly, this is the case when tomographic retrievals are used. Are you using the tomographic retrieval scheme? I did not find this information in your paper.

A major part of this work is to compare the GLORIA measurements with the models ICON and EMAC. You write several time, that even the fine structures are remarkably well resolved.

We thank the Editor for pointing out this weakness of our discussion and agree that clarification is required. A tomographic retrieval scheme was not used in our study. We applied conventional 1-D retrievals of single vertical profiles. Profiles are combined to 2-D vertical cross sections along flight track (see Johansson et al., 2018a). For clarification, we modified P5/L9-10 as follows:

"Details on the applied 1-D trace gas retrieval and the data products used here are provided by Johansson et al. (2018a). The retrieved individual trace gas profiles of GLORIA are combined to 2-D vertical cross sections of the respective species along the flight track."

The retrievals can be characterised with respect to (i) vertical smoothing by a conventional 1-D averaging kernel and (ii) horizontal smoothing along the viewing direction (i.e. to the right hand side of the aircraft) by a more complex 2-D averaging kernel. The latter is discussed by Ungermann et al. (2011) in their Sect 3.2 and Eqn (9) (although these authors focus at tomographic retrievals). The application of the 2-D averaging kernel specific to GLORIA chemistry mode observations (such as used here) is discussed by Woiwode et al. (2018). We added this reference in the manuscript.

In the vertical domain, conventional 1-D averaging kernels from the trace gas retrievals describe the amount of smoothing, which results from the constraint (in our case a Tikhonov approach), which reduces the vertical resolution. However, the achieved resolution in the altitude domain used for the respective trace gases is here in the order of 500 m and therefore well comparable to the vertical resolution of the models. Therefore, the application of conventional 1-D averaging kernels to the model data in the vertical domain is not expected to affect the comparison notably and the direct interpolation of model data to the GLORIA tangent points, such as done in many other studies (see e.g. Johansson et al., 2019; Khosrawi et al., 2017; Braun et al., 2019), is applied here. The quality of the GLORIA data used here is furthermore confirmed by in situ comparisons. As discussed by Johansson et al. (2018a), median

differences and median absolute deviations between the GLORIA chemistry mode data and in situ observations during the entire PGS campaign were as low as  $-0.13 \pm 0.63$  ppmv (water vapour),  $-3.5 \pm 116.8$  ppbv (ozone), and  $-0.03 \pm 0.85$  ppbv (nitric acid). Since the in situ data measured during PGS covered regions characterised by strong vertical gradients, this gives us confidence that the GLORIA data are not affected by overall systematic biases that are relevant here.

In the horizontal domain along the viewing direction (i.e. to the right side of the flight track), model comparisons can be affected by limited horizontal resolution of a limb sounder along the line of sight. This effect can be taken into account by applying a 2-D averaging kernel for the interpolation of the model data, which takes into account also the dimension along viewing direction (Ungermann et al., 2011). However, this approach is computationally demanding, and leads, if strong gradients along the line of sight are avoided, only to a moderate improvement of direct comparisons of fine structures in vertical cross sections. This aspect has been investigated by Woiwode et al. (2018) (see their Appendix A) for the case of water vapour, when detailed observations of a tropopause fold were compared with high resolution ECMWF IFS data. In contrast, in case of strong local variations and gradients along the line of sight, as in the case of temperature variations due to gravity waves (which are not analysed here), the application of 2-D averaging kernels can strongly improve direct comparisons.

The flight analysed here was planned with the help of forecasts so that strong gradients in the meteorological fields along the line of sight were mostly avoided. The GLORIA observations were mostly aligned in a way that the viewing direction was aligned into relatively homogeneous air masses along viewing direction (compare Fig. 1b with 4b). This enabled us to resolve the fine filaments discussed in Section 4.3 that agree remarkably well with the model data. Naturally, an optimal alignment is not possible in all cases, and local differences between the GLORIA and model cross sections of trace gases due to remaining effects by horizontal gradients cannot be excluded. However, when the GLORIA and model data are analysed as ensemble (e.g. in mean values of correlations), these remaining smoothing effects are expected to cancel out on average in the large amount of data used here.

**For clarification, we modified P10/L1-8 as follows:**

"The vertical resolution of the GLORIA data used here is in the order of 500 m, depending on altitude and parameter (see Johansson et al., 2018a), and therefore comparable with the vertical resolution of the simulations by both models in the tropopause region. Therefore, the use of 1-D averaging kernels in the vertical domain, such as often used in context of vertical profiles retrieved from satellite limb observations (e.g. Microwave Limb Sounder (MLS)) that are characterized by notably coarser vertical resolution is not expected to improve the comparison significantly. The absence of relevant overall systematic biases in the GLORIA data used here is furthermore confirmed by in situ comparisons (see Johansson et al., 2018a).

Due to the limb viewing geometry, strong horizontal gradients along the line of sight of GLORIA (i.e. towards the right hand side of the flight track) can affect direct comparisons of vertical cross sections of atmospheric parameters derived from the GLORIA observations and interpolated from the models at the tangent points. This effect can be taken into account by interpolating the model data with the help of 2-D averaging kernels (Ungermann et al., 2011, their Sect. 3.2). As discussed by Woiwode et al. (2018) in a case study where the mesoscale fine structure of a tropopause fold was investigated, the application of 2-D averaging kernels improves the model comparison only moderately if the observations are aligned such that horizontal gradients in the trace gas fields along the line of sight are small (see their Appendix A).

Aided by meteorological forecasts, the flight analysed here was planned so that the GLORIA observations were mostly aligned in such a way. This can be seen by comparing Fig. 1b with Fig. 4b, for example during the backward leg to Kiruna, when the GLORIA limb views were aligned along the

direction of moist filaments above Greenland. Therefore, the viewing geometry allowed us to resolve the fine structures of the narrow filaments discussed in Sect. 4.3 remarkably well. Due to the suitable alignment of the GLORIA observations during the discussed flight and since the application of 2-D averaging kernels is computationally demanding (particularly in case of the GLORIA high spectral resolution chemistry mode observations that employ a large number of spectral sampling points), 2-D averaging kernels are not applied here. Therefore, local discrepancies between the GLORIA and model cross sections due to remaining effects by horizontal gradients along the line of sight cannot be excluded.

However, when the complete ensemble of GLORIA and model data points is analysed, such remaining effects by horizontal gradients are expected to cancel out on average due to the large amount of data points. Therefore, we consider the estimation of model biases in Sect. 4.4 to be robust."

On page 16 lines 27 to 30 you find a mismatch and argue, that this can be explained by 'line of sight effects in the GLORIA observations'. But at page 10, line 1 - 8 you write line of sight effects are expected to cancel out due to the large amount of data (is this because you use a tomographic retrieval?).

We thank the Editor for pointing out the unclear discussion. On P10/L1–8 we meant to refer to the trace gas retrievals and the discussed correlations with the respective trace gases in the models. As mentioned above, improvements of local structures in the vertical cross section comparisons are possible, if 2-D averaging kernels were applied to the model results. However, when looking at the whole ensemble of data from the flight here, involving all kinds of viewing directions, we expect remaining line of sight effects to cancel out, when we analyse the model biases.

However, when looking at the distributions of clouds in the 2-D cross sections along the flight track, as described on page 16, line of sight effects could explain discrepancies between models and GLORIA observations. Note that, different from the trace gases, the cloud signal represented by the cloud index (CI) is directly inferred from the GLORIA spectral measurements, not by a retrieval procedure. While GLORIA measurements show an integrated cloud signal along the line of sight, the model data has been interpolated to the tangent point of the GLORIA measurements, which represents one single point in space. If, for instance, clouds were situated in front or behind the tangent point along the line of sight, this comparison would lead to a discrepancy between the model results and the measurements. In addition, the properties of clouds with their strong optical gradients (transparent/opaque) are considerably different from the trace gas distributions. Especially, complex small-scale cloud structures can differ in coverage and orientation from the trace gas fields. Therefore, we consider the comparison of GLORIA cloud detection to the simulated clouds to be more difficult, in particular for small clouds or edges of clouds. Despite these limitations of the comparison, we mostly found good agreement between GLORIA and the models.

For clarification, we added after P16/L27:

"Note that line of sight related effects are capable of particularly strong influences on the comparison with respect to clouds. If, for instance, clouds were situated in front or behind the tangent point along the line of sight, this comparison would lead to a discrepancy between the model results and the measurements. Especially, complex small-scale cloud structures with strong optical gradients (transparent/opaque) can differ in coverage and orientation when compared to the trace gas fields. Therefore, we consider the comparison of GLORIA cloud detection to the simulated clouds to be more

**difficult, in particular for small clouds or edges of clouds. Despite these limitations of the comparison, we mostly found good agreement between GLORIA and the models."**

On page 19 line 23 - 32 you describe biases around the tropopause. Especially in regions with large gradients AVK effects may cause under- and over-estimations similar to the describes effects.

In my view the AVK issue leaves a gap in your paper, because it is not clear, if discrepancies in some cases are due to the neglect of resolution effects, which would disappear if AVK's would be used.

See discussion above: In the vertical domain, the vertical resolution of the GLORIA retrieval results is comparable with the vertical model resolution and the absence of relevant overall systematic biases in the GLORIA data is confirmed by in situ comparisons. In the horizontal domain, the use of 2-D averaging kernels is capable to improve the comparison in the presence of strong horizontal gradients along the line of sight. However, since such gradients were mostly avoided during the discussed flight and since the application is demanding, 2-D averaging kernels were not applied here. However, when the complete ensemble of GLORIA and model data points is analysed, such remaining effects by horizontal gradients are expected to cancel out on average due to the large amount of data points. Therefore, we consider the estimation of model biases in Sect. 4.4 to be robust.

Some language issues. I am not an expert in English, so I am not entirely sure, if my corrections are entirely correct. I also may have missed other errors.

Page 11 line 9 accompanied by with a notably positive NAO -> accompanied by a notably positive NAO

Done

Page 26 line 16 than in the case of \_THE\_ ICON-ART forecast

Done

Page 28 line 5 in significant low biases -> in significantly lower biases

For clarification, we rephrased to "in significantly lower mixing ratios"

Page 29 line 6 by tuning of this scheme -> by tuning this scheme

Done

**References**

Woiwode, W., Dörnbrack, A., Bramberger, M., Friedl-Vallon, F., Haenel, F., Höpfner, M., Johansson, S., Kretschmer, E., Krisch, I., Latzko, T., Oelhaf, H., Orphal, J., Preusse, P., Sinnhuber, B.-M., and Ungermann, J.: Mesoscale fine structure of a tropopause fold over mountains, Atmos. Chem. Phys., 18, 15643–15667, https://doi.org/10.5194/acp-18-15643-2018, 2018.